# THE MULTI-BLOCK DC FUNCTION CLASS: THEORY, ALGORITHMS, AND APPLICATIONS

## ABSTRACT

We present the Multi-Block DC (BDC) class, a broad class of structured nonconvex functions that admit a DC ("difference-of-convex") decomposition across parameter blocks. This block structure not only subsumes the usual DC programming, it turns out to be provably more powerful. Specifically, we demonstrate how standard models (e.g., polynomials and tensor factorization) *must have* DC decompositions of exponential size, while their BDC formulation is polynomial. This separation in complexity also underscores another *key aspect*: unlike DC formulations, obtaining BDC formulations for problems is vastly easier and constructive. We illustrate this aspect by presenting explicit BDC formulations for modern tasks such as deep ReLU networks, a result with no known equivalent in the DC class. Moreover, we complement the theory by developing algorithms with non-asymptotic convergence theory, including both batch and stochastic settings, and demonstrate the broad applicability of our method through several applications.

## 1 INTRODUCTION

The growing complexity of machine learning raises numerous challenges for nonconvex optimization, of which the identification of problem formulations that model, expose, and exploit structure is of key importance. A specific example of this idea is the class of difference-of-convex (DC) functions, which captures problem structure that has not only been well-studied over the decades but has attracted significant attention recently (Khamaru and Wainwright, 2018; Davis et al., 2022; Maskan et al., 2025). However, identifying a tractable and practically useful DC decomposition for a given problem can be difficult, even NP-hard in some cases (Ahmadi and Hall, 2018). Although the existence of DC decompositions can be guaranteed under mild smoothness assumptions (Tuy, 2016), these classical results are often non-constructive and offer no guidance on how to obtain a suitable decomposition.

These challenges call for a more flexible perspective, so rather than insisting on a single global DC decomposition we advocate a shift toward multi-block DC structure. Formally, we consider the minimization of a function $f(\theta_1, \ldots, \theta_n)$ that admits a DC decomposition with respect to each block $\theta_i$ individually, when all other blocks are fixed. To our knowledge, this perspective has been explored only in the two-block case, and surprisingly, remains largely unstudied in the multi-block setting, despite being highly amenable to practice. We show how multi-block DC decompositions are easy to construct, align naturally with the structure of many modern machine learning problems, and admit algorithms with convergence guarantees comparable to the classical DC framework.

In light of the above motivation, we summarize our main contributions as follows:

- We define a new class called *multi-block* DC functions (hereafter, BDC), which extends the so-called partial DC framework from two-blocks to multiple blocks. We study fundamental properties of this class, including an exponential separation in representation complexity compared to DC formulations. We present both examples and concrete tools to show how one can flexibly formulate problems to be BDC, underscoring the class's practicality across several applications.

- We propose a multi-block variant of the DC algorithm designed for BDC functions, which exploits the block structure to perform efficient updates. We further extend this algorithm to a stochastic setting, where the decomposition functions are accessed only through noisy oracles, broadening the practical applicability of our framework to large-scale machine learning problems.

## 2 PROBLEM SETUP AND RELATED WORK

We consider the general optimization problem

$$\min_{\boldsymbol{\theta} \in \mathcal{X}} f(\boldsymbol{\theta}), \tag{2.1}$$

where $f : \mathcal{X} \to \mathbb{R}$ is possibly nonconvex and $\mathcal{X} \subseteq \mathbb{R}^d$ is the domain of our objective.

**Block structure.** We assume that $\mathcal{X}$ admits a Cartesian product decomposition $\mathcal{X} = \mathcal{X}_1 \times \cdots \times \mathcal{X}_n$, where each $\mathcal{X}_i \subseteq \mathbb{R}^{d_i}$ and $d = \sum_{i=1}^n d_i$. We also define $\bar{\mathcal{X}}_i = \mathcal{X}_1 \times \cdots \times \{0\}^{d_i} \times \cdots \times \mathcal{X}_n$, the set obtained from $\mathcal{X}$ by forcing the $i$th block to be zero, for notational convenience.

Let $D_i \in \mathbb{R}^{d_i \times d}$ be the selection matrix that extracts the $i$th block of $\boldsymbol{\theta}$. Equivalently, $D_i$ is obtained by taking $d_i$ distinct rows of the identity matrix $I_d$, so that $\{D_i\}_{i=1}^n$ forms a non-overlapping partition of the coordinates and satisfies $\sum_{i=1}^n D_i^\top D_i = I_d$.

For clarity, we use boldface letters (e.g., $\boldsymbol{\theta} \in \mathcal{X}$) to denote full decision variables, and non-boldface symbols (e.g., $\theta_i \in \mathcal{X}_i$) to denote individual blocks. We define, for each $i \in [n]$, $\theta_i := D_i \boldsymbol{\theta}$, so that $\theta_i \in \mathbb{R}^{d_i}$ represents the $i$th block of $\boldsymbol{\theta}$. We also define the block-extended vector $\boldsymbol{\theta}_i := D_i^\top D_i \boldsymbol{\theta}$, which coincides with $\boldsymbol{\theta}$ on block $i$ and is zero elsewhere. Its complement is $\bar{\boldsymbol{\theta}}_i := (I_d - D_i^\top D_i)\boldsymbol{\theta}$, so that $\boldsymbol{\theta} = \boldsymbol{\theta}_i + \bar{\boldsymbol{\theta}}_i$. These definitions will be useful for expressing multi-block DC decompositions and coordinate updates. We are now ready to state our key structural assumption on $f$ and establish closure properties of the induced function class.

**Assumption 1** (Multi-Block DC separability). We assume that $f : \mathcal{X} \to \mathbb{R}$ admits a DC decomposition with respect to each block $\theta_i$ when all other blocks are fixed. Formally, this means that for each $i \in [n]$, there exist functions $g_i, h_i : \mathcal{X}_i \times \bar{\mathcal{X}}_i \to \mathbb{R}$, such that for every $\boldsymbol{\theta} \in \mathcal{X}$,

$$f(\boldsymbol{\theta}) = g_i(\theta_i; \bar{\boldsymbol{\theta}}_i) - h_i(\theta_i; \bar{\boldsymbol{\theta}}_i),$$

where $g_i(\cdot\,; \bar{\boldsymbol{\theta}}_i)$ and $h_i(\cdot\,; \bar{\boldsymbol{\theta}}_i)$ are convex in $\theta_i$. We refer to this property as a BDC decomposition.

**Proposition 2.1** (Closure). *Let $f_i$ be* BDC *functions for $i = 1, \ldots, m$, Then, the following functions are also* BDC*:    (i) $\sum_{i=1}^m \alpha_i f_i$, for $\alpha_i \in \mathbb{R}$,    (ii) $\min_{i=1,\ldots m} f_i$,    (iii) $\max_{i=1,\ldots m} f_i$*

Proof can be found in Appendix B.1. It is worth noting that the class of BDC functions is strictly larger than the classical DC family. For instance, Veselý and Zajíček (2018) construct a function in $\mathbb{R}^2$ that is DC on every convex curve but does not admit a global DC decomposition, implying it is BDC but not DC. The main appeal of the BDC class, however, is not merely its greater expressiveness but its flexibility. In practice, BDC decompositions are easier to construct than global DC decompositions, and in many cases of practical interest they can be obtained explicitly in a constructive manner.

**Example** (Tensor decomposition). Let $\mathcal{T} \in \mathbb{R}^{m_1 \times \cdots \times m_n}$ be an $n$th-order tensor, and let $\theta_i \in \mathbb{R}^{m_i \times r}$ denote the $i$th factor matrix. The canonical polyadic (CP) decomposition solves

$$\min_{\theta_1, \ldots, \theta_n} \frac{1}{2} \big\| \mathcal{T} - [\![\theta_1, \ldots, \theta_n]\!] \big\|_F^2,$$

where $[\![\theta_1, \ldots, \theta_n]\!]$ denotes the rank-$r$ CP reconstruction. This problem is nonconvex jointly in all $\theta_i$'s, but convex in each $\theta_i$ when the others are fixed. This gives a BDC decomposition with $h_i = 0$, which also underlies the classical alternating least-squares (ALS) algorithm, which performs exact multi-block minimization steps. Although this is a purely multi-block convex structure ($h_i = 0$), it illustrates how BDC decompositions are easier to obtain than DC decompositions, which would be algebraically complex in this case. We present more general examples with nontrivial $h_i$ in Section 3.

### 2.1 RELATED WORK

DC programming has been employed in a wide range of machine learning applications from kernel selection (Argyriou et al., 2006) to discrepancy estimation for domain adaptation (Awasthi et al., 2024). The classical method for solving DC problems is the DC Algorithm (DCA), introduced by Tao and Souad (1986). The first asymptotic convergence results for DCA were established by Tao (1997), with a simplified analysis under differentiability assumptions later provided by Lanckriet and Sriperumbudur (2009). More recently, non-asymptotic convergence rates of $\mathcal{O}(1/k)$ have been

established (Khamaru and Wainwright, 2018; Yurtsever and Sra, 2022; Abbaszadehpeivasti et al., 2023). For a comprehensive survey, we refer to (Le Thi and Pham Dinh, 2018; 2024).

Despite its generality, the class of BDC functions remains largely unexplored. The only prior study we are aware of is (Pham Dinh et al., 2022) that considers only the two-block case (termed partial DC decomposition) and proposes the Alternating DC algorithm. Their method converges to weak critical points in general, and to Fréchet/Clarke critical points under the Kurdyka–Łojasiewicz property, with numerical validation on a nonconvex feasibility problem (intersection of two nonconvex sets) and robust PCA. However, their results investigate neither the constructive structure (algebra) of BDC functions nor their broader application potential, topics that we address through a general multi-block formulation and algorithms with non-asymptotic convergence guarantees.

Finally, our framework should not be confused with the block-coordinate DCA of Maskan et al. (2024), which tackles the simpler classical DC problem with a fixed global decomposition and develops a block-coordinate algorithm. In contrast, we introduce and study the BDC problem class, yielding a broader and much more flexible formulation. Our work, moreover, calls for a conceptual shift: rather than seeking a global DC decomposition, we advocate a multi-block decomposition, as this is vastly easier to construct, more expressive, and often algorithmically advantageous.

## 3 WHY THE BDC FUNCTION CLASS?

We discuss two important types of functions to motivate the BDC class. First, we prove that the complexity of a DC decomposition for monomials is exponentially higher than its BDC counterpart. Second, we propose an explicit BDC decomposition for deep ReLU networks (their architectural core), which we then expand to cover regression and classification tasks.

### 3.1 DC AND BDC COMPLEXITY OF A MONOMIAL

Let $\boldsymbol{\theta} = (\theta_1, \cdots, \theta_n)$ and $f(\boldsymbol{\theta}) = \theta_1^{b_1} \theta_2^{b_2} \cdots \theta_n^{b_n}$ with $s = \sum_{i=1}^{n} b_i$. We measure DC and BDC complexities by the minimum number of atoms needed to represent a decomposition. For the DC class, take $f(\boldsymbol{\theta}) = g(\boldsymbol{\theta}) - h(\boldsymbol{\theta})$ with $g(\boldsymbol{\theta}) = \sum_{i=1}^{r} \alpha_i \phi_i(\boldsymbol{\theta})$ and $h(\boldsymbol{\theta}) = \sum_{i=r+1}^{r+q} \alpha_i \phi_i(\boldsymbol{\theta})$, where each $\alpha_i > 0$ and $\phi_i$ is a convex atom: $\phi_i(\boldsymbol{\theta}) = (u_i^\top \boldsymbol{\theta})^s$ if $s$ is even, and $\phi_i(\boldsymbol{\theta}) = (u_i^\top \boldsymbol{\theta} + d_i)^{s+1}$ if $s$ is odd. We denote by $N$ the *minimum atom count*, i.e., the minimum of $r + q$ over all such decompositions. Using the notion of *Waring rank* (Carlini et al., 2012) and the *polarization property* (Drápal and Vojtěchovský, 2009), we bound $N$ in the following Theorem 3.1. The detailed proof and definitions needed for this result are given in Appendix B.2.

**Theorem 3.1** (DC complexity for monomials). *Consider* $f(\boldsymbol{\theta}) = \prod_{i=1}^{n} \theta_i^{b_i}$ *with* $1 \le b_1 \le \cdots \le b_n$ *and* $s = \sum_i b_i$. *Then the minimum atom count* $N$ *for* DC *decomposition is either of the following:*

- *If* $s$ *is even and atoms are of the form* $(u^\top \boldsymbol{\theta})^s$, *then* $\prod_{i=2}^{n}(b_i + 1) \le N \le \left\lfloor \frac{1}{2} \prod_{i=1}^{n}(b_i + 1) \right\rfloor$.

- *If* $s$ *is odd and atoms are of the form* $(u^\top \boldsymbol{\theta} + d)^{s+1}$, *then* $N = \prod_{i=1}^{n}(b_i + 1)$.

As Theorem 3.1 shows, the DC decomposition of a monomial requires a very large number of atoms. In contrast, a BDC decomposition can be significantly more compact. In the simplest case, each $\theta_i^{b_i}$ is treated as a standalone block, reducing the atom count *exponentially* compared to the DC decomposition. More generally, one may split the monomial into a few larger blocks, decompose each block, and then multiply the resulting sums, thereby reducing the complexity. For instance, the monomial $\theta_1 \theta_2 \theta_3^2 \theta_4^4$ requires at least 30 atoms in a DC representation, which matches the lower bound of Theorem 3.1. Instead taking the trivial blocks $\theta_1$, $\theta_2$, $\theta_3^2$, and $\theta_4^4$, yields a BDC decomposition with only 4 atoms. Alternatively, splitting into two blocks, $\theta_1 \theta_2$ and $\theta_3^2 \theta_4^4$, results in $2 + 7 = 9$ atoms in total through Theorem 3.1. An explicit BDC decomposition in this case is

$$\theta_1 \theta_2 \theta_3^2 \theta_4^4 = \frac{1}{14400} \left[ (\theta_1 + \theta_2)^2 - (\theta_1 - \theta_2)^2 \right] \times \left[ 5 \left( (\theta_3 + \theta_4)^6 + (\theta_3 - \theta_4)^6 \right) \right.$$

$$\left. + 3 \left( (\theta_3 + 3\theta_4)^6 + (\theta_3 - 3\theta_4)^6 \right) - 8 \left( (\theta_3 + 2\theta_4)^6 + (\theta_3 - 2\theta_4)^6 + 420\theta_4^6 \right) \right].$$

## 3.2 BDC FORMULATION OF A DEEP RELU NETWORK

Consider an $L$-layer ReLU network parameterized by $\boldsymbol{\theta} = \big(W_1, b_1, \ldots, W_L, b_L\big)$. For input $x \in \mathbb{R}^d$, define $a_0(x) = x$, and

$$F_x(\boldsymbol{\theta}) = W_L a_{L-1}(x) + b_L, \quad a_l(x) = \sigma\big(W_l\, a_{l-1}(x) + b_l\big), \quad l = 1, \ldots, L-1,$$

where $W_l$ are weight matrices, $b_l$ are bias vectors, and $\sigma(\cdot)$ denotes the ReLU activation. Here $W_L \in \mathbb{R}^{C \times d_L}$ and $b_L \in \mathbb{R}^C$ represent the weights of the output layer. For regression, we take $C = 1$; for classification, $C$ is the number of classes.

Now, we aim to express the network output as a BDC function in each class. We begin by writing each activation using two nonnegative multi-block component-wise convex functions $a_l = Z_l^+ - Z_l^-$, with the following initialization and forward recursion:

***Initialization** ($l = 1$):*     $Z_1^+ = \sigma(W_1 x + b_1), \qquad Z_1^- = 0.$

***Forward recursion** ($l \to l+1$):* given $(Z_l^+, Z_l^-)$ with $a_l = Z_l^+ - Z_l^-$, define

$$p_{l+1} = \sigma(W_{l+1})\, Z_l^+ + \sigma(-W_{l+1})\, Z_l^- + b_{l+1},$$

$$Z_{l+1}^- = \sigma(W_{l+1})\, Z_l^- + \sigma(-W_{l+1})\, Z_l^+, \qquad Z_{l+1}^+ = \max\{p_{l+1}, Z_{l+1}^-\}.$$

Using $\sigma(a - b) = \max\{a, b\} - b$, we obtain

$$Z_{l+1}^+ - Z_{l+1}^- = \sigma\big(W_{l+1}(Z_l^+ - Z_l^-) + b_{l+1}\big) = \sigma(W_{l+1} a_l + b_{l+1}) \ = \ a_{l+1}(x).$$

This recursion guarantees $Z_l^{\pm} \geq 0$ and that each component of $Z_l^{\pm}$ is convex in the chosen block $\theta_l = (W_l, b_l)$; the used operations (nonnegative linear maps and coordinatewise maxima) preserve convexity and nonnegativity layer by layer.

***Output layer:*** Define nonnegative functions

$$\begin{aligned}
A(\boldsymbol{\theta}) &:= \sigma(W_L) Z_{L-1}^+ + \sigma(-W_L) Z_{L-1}^- + \sigma(b_L), \\
B(\boldsymbol{\theta}) &:= \sigma(W_L) Z_{L-1}^- + \sigma(-W_L) Z_{L-1}^+ + \sigma(-b_L).
\end{aligned} \tag{3.1}$$

Then, $F_x(\boldsymbol{\theta}) = A(\boldsymbol{\theta}) - B(\boldsymbol{\theta})$. The following Theorem 3.2 proves that each component of $A(\boldsymbol{\theta})$ and $B(\boldsymbol{\theta})$ in (3.1) is a convex function in every block (See Appendix B.3 for the proof).

**Theorem 3.2** (Validity of BDC decomposition for Deep ReLU Network). *For any block $\theta_l = (W_l, b_l)$, (3.1) gives $A(\boldsymbol{\theta})$ and $B(\boldsymbol{\theta})$ such that each component of $A(\cdot\,; \bar{\boldsymbol{\theta}}_l)$ and $B(\cdot\,; \bar{\boldsymbol{\theta}}_l)$ is nonnegative and convex in $\theta_l$, and we have $F_x(\boldsymbol{\theta}) = A(\boldsymbol{\theta}) - B(\boldsymbol{\theta})$.*

Our result in Theorem 3.2 provides an explicit BDC formulation for deep ReLU networks. While it is known (as an existence result) that deep ReLU networks are DC, explicit DC decompositions are currently available only for *shallow* networks (Askarizadeh et al., 2024).

### 3.2.1 REGRESSION WITH MSE LOSS: BDC FORMULATION

For a label $y \in \mathbb{R}$ and scalar output $F_x(\boldsymbol{\theta}) = A(\boldsymbol{\theta}) - B(\boldsymbol{\theta})$, the Mean Squared Error (MSE) loss is $\mathcal{L}_{x,y}^{\mathrm{MSE}}(\boldsymbol{\theta}) := \big(F_x(\boldsymbol{\theta}) - y\big)^2$. This yields the explicit BDC decomposition

$$\mathcal{L}_{x,y}^{\mathrm{MSE}}(\boldsymbol{\theta}) = 2\big(A^2(\boldsymbol{\theta}) + (B(\boldsymbol{\theta}) + y)^2\big) - (A(\boldsymbol{\theta}) + B(\boldsymbol{\theta}) + y)^2, \tag{3.2}$$

a difference of two multi-block convex functions if $y \geq 0$.

**Remark 3.3.** *If labels $y$ are not guaranteed to be nonnegative, one can shift labels and outputs by a constant $c \geq 0$ so that $y + c \geq 0$. This translation does not affect the BDC structure, so the assumption $y \geq 0$ is not restrictive.*

**Correctness.** By Theorem 3.2, $A(\boldsymbol{\theta}), B(\boldsymbol{\theta}) \geq 0$ are multi-block convex. For $y \geq 0$ we have $B(\boldsymbol{\theta}) + y \geq 0$ and $A(\boldsymbol{\theta}) + B(\boldsymbol{\theta}) + y \geq 0$, so $A^2(\boldsymbol{\theta})$, $(B(\boldsymbol{\theta}) + y)^2$, and $(A(\boldsymbol{\theta}) + B(\boldsymbol{\theta}) + y)^2$ are multi-block convex (square is convex and nondecreasing on $[0, \infty)$). Therefore (3.2) gives a valid BDC decomposition of $\mathcal{L}_{x,y}^{\mathrm{MSE}}(\boldsymbol{\theta})$.

### 3.2.2 CLASSIFICATION WITH CE LOSS: BDC FORMULATION

Before we can establish a BDC formulation of the Cross-Entropy (CE) loss, we need a general result that extends BDC decompositions to more complex structures. Specifically, we develop a composition principle ensuring that when the input admits a BDC decomposition, the expression obtained through a conjugate function can also be written explicitly in BDC form. The following Proposition 3.4 establishes this principle (see Appendix B.4 for the proof). In contrast to many DC composition rules that only guarantee existence, this result is *constructive*.

**Proposition 3.4** (BDC decomposition for $f^* \circ E$)**.** *Let $U \subset \mathbb{R}^m$ be compact, $f : U \to \mathbb{R}$ finite, and $f^*(t) = \max_{u \in U} \{\langle u, t \rangle - f(u)\}$ be the conjugate of $f$. Suppose $E(\boldsymbol{\theta}) = (E_1(\boldsymbol{\theta}), \ldots, E_m(\boldsymbol{\theta}))$, where each component $E_j$ is BDC, i.e., $E_j(\boldsymbol{\theta}) = a_{ij}(\theta_i; \bar{\boldsymbol{\theta}}_i) - b_{ij}(\theta_i; \bar{\boldsymbol{\theta}}_i)$ for every block $i \in [n]$. For $j = 1, \ldots, m$ set $\underline{u}_j := \min_{u \in U} u_j$, $\bar{u}_j := \max_{u \in U} u_j$, $c_j^+ := \max\{-\underline{u}_j, 0\}$, $d_j^+ := \max\{\bar{u}_j, 0\}$. Define the vectors $c^+$, $d^+ \in \mathbb{R}^m$. Then $f^* \circ E$ is BDC, with an explicit multi-block decomposition $f^*(E(\boldsymbol{\theta})) = g_i(\theta_i; \bar{\boldsymbol{\theta}}_i) - h_i(\theta_i; \bar{\boldsymbol{\theta}}_i)$, where, for each block $i$,*

$$h_i(\theta_i; \bar{\boldsymbol{\theta}}_i) := \langle c^+, a_i(\theta_i; \bar{\boldsymbol{\theta}}_i) \rangle + \langle d^+, b_i(\theta_i; \bar{\boldsymbol{\theta}}_i) \rangle,$$

$$g_i(\theta_i; \bar{\boldsymbol{\theta}}_i) := f^*(E(\boldsymbol{\theta})) + h_i(\theta_i; \bar{\boldsymbol{\theta}}_i),$$

*with $a_i(\theta_i; \bar{\boldsymbol{\theta}}_i) := (a_{i1}(\theta_i; \bar{\boldsymbol{\theta}}_i), \ldots, a_{im}(\theta_i; \bar{\boldsymbol{\theta}}_i))$, $b_i(\theta_i; \bar{\boldsymbol{\theta}}_i) := (b_{i1}(\theta_i; \bar{\boldsymbol{\theta}}_i), \ldots, b_{im}(\theta_i; \bar{\boldsymbol{\theta}}_i))$.*

Using the split $F_x(\boldsymbol{\theta}) = A(\boldsymbol{\theta}) - B(\boldsymbol{\theta})$ in (3.1), for a label $y \in \{1, \ldots, C\}$ the CE loss is $\mathcal{L}_{x,y}^{\mathrm{CE}}(\boldsymbol{\theta}) = \mathrm{LSE}(F_x(\boldsymbol{\theta})) - A_y(\boldsymbol{\theta}) + B_y(\boldsymbol{\theta})$, where $\mathrm{LSE}(\cdot)$ is the log-sum-exp with variational form

$$\mathrm{LSE}(t) = \max_{p \in \Delta_C} \{\langle p, t \rangle - \mathrm{Ent}(p)\}, \quad \Delta_C := \{p \geq 0, \ \mathbf{1}^\top p = 1\}, \quad \mathrm{Ent}(p) := \sum_{c=1}^C p_c \log p_c.$$

**Corollary 3.5.** *Applying Proposition 3.4 with $U = \Delta_C$ and $f = \mathrm{Ent}$ yields $\underline{u}_j = 0$ and $\bar{u}_j = 1$ for all $j$, hence $c^+ = 0$ and $d^+ = 1$. Therefore, for every $x, y$, $\mathcal{L}_{x,y}^{\mathrm{CE}}(\boldsymbol{\theta}) = g(\boldsymbol{\theta}) - h(\boldsymbol{\theta})$, where*

$$g(\boldsymbol{\theta}) := \mathrm{LSE}(F_x(\boldsymbol{\theta})) + \mathbf{1}^\top B(\boldsymbol{\theta}) + B_y(\boldsymbol{\theta}), \qquad h(\boldsymbol{\theta}) := A_y(\boldsymbol{\theta}) + \mathbf{1}^\top B(\boldsymbol{\theta}).$$

**Correctness.** For any parameter block, by Theorem 3.2 each component of $A(\boldsymbol{\theta})$ and $B(\boldsymbol{\theta})$ is convex. Convexity of $g(\cdot \, ; \bar{\boldsymbol{\theta}}_l)$ and $h(\cdot \, ; \bar{\boldsymbol{\theta}}_l)$ in block $l$ follows directly from Proposition 3.4 with shifts $c^+ = 0$ and $d^+ = 1$. Therefore $\mathcal{L}_{x,y}^{\mathrm{CE}}(\boldsymbol{\theta})$ is a valid BDC function.

## 4 BDC ALGORITHM

In this section we propose BDC algorithms (BDCA) along with their convergence results for BDC optimization (2.1) under assumptions of $L$-smoothness, generalized smoothness, and stochasticity. Unlike the conventional DCA, our BDC algorithm considers a convex surrogate function obtained by linearizing the concave component of the objective function on each randomly chosen block $i_k$ around the update point, $\boldsymbol{\theta}^k$, at $k^{\mathrm{th}}$ iteration. Throughout this section we denote $\mathcal{G}(\boldsymbol{\theta}) := \sup_{u \in \partial f(\boldsymbol{\theta})} \|u\|$.

### 4.1 BDCA UNDER $L$-SMOOTHNESS

Assume BDC problem (2.1), when each $g_i(\theta_i; \bar{\boldsymbol{\theta}}_i)$ satisfies $L_i$-smoothness and $L := \max_{i \in [n]} L_i$. Our BDC algorithm at $k^{\mathrm{th}}$ iteration will select a block $i_k$ uniformly at random, and then update by minimizing a surrogate function on the selected block, as:

$$\theta_{i_k}^{k+1} \in \underset{\theta_{i_k} \in \mathcal{X}_{i_k}}{\mathrm{argmin}} \, g_{i_k}(\theta_{i_k}; \bar{\boldsymbol{\theta}}_{i_k}^k) - \langle u_{i_k}^k, \theta_{i_k} \rangle, \tag{4.1}$$

where $u_{i_k}^k \in \partial h_{i_k}(\theta_{i_k}^k; \bar{\boldsymbol{\theta}}_{i_k}^k)$. After solving (4.1), we update $\boldsymbol{\theta}^{k+1} = \bar{\boldsymbol{\theta}}_{i_k}^k + \theta_{i_k}^{k+1}$, and set $k = k + 1$. The convergence guarantee for (4.1) is summarized in the following corollary. When the problem has convex and compact constraints, we propose a more general convergence result in Appendix A.1.

**Corollary 4.1.** *The sequence generated by the update (4.1) will satisfy*

$$\min_{k \in \{1, \ldots, K\}} \mathbb{E}_i \left[ \mathcal{G}^2(\boldsymbol{\theta}^k) \right] \leq \frac{2Ln}{K} \left( f(\boldsymbol{\theta}^1) - f^\star \right), \tag{4.2}$$

*where $\mathbb{E}_i[.]$ denotes expectation w.r.t. the $i^{th}$ block choice.*

## 4.2 PROXIMAL BDCA UNDER GENERALIZED SMOOTHNESS ASSUMPTION

Many optimization objectives do not possess a Lipschitz continuous gradient. Despite this, recent studies have shown that in some important training tasks a more relaxed smoothness assumption holds (Zhang et al., 2019; Crawshaw et al., 2022). This assumption essentially bounds the norm of the Hessian of the objective with a function of the gradient norm. Motivated by this, we conducted a simulation showing that a multi-block reminiscent of the generalized smoothness holds when training a neural network (see Figure 1). Based on these observations, we assume a more relaxed assumption on the components of $g(\boldsymbol{\theta})$, known as $\ell$-smoothness defined below.

**Definition 1** ($\ell$-smoothness,(Li et al., 2024)). A real-valued differentiable function $g_i : \mathcal{X}_i \times \bar{\mathcal{X}}_i \to \mathbb{R}$ is $\ell$-smooth for continuous function $\ell : [0, +\infty) \to (0, +\infty)$ where $\ell$ is non-decreasing, if it satisfies $\|\nabla^2 g_i(\theta_i; \bar{\boldsymbol{\theta}}_i)\| \leq \ell(\|\nabla g_i(\theta_i; \bar{\boldsymbol{\theta}}_i)\|)$ for fixed $\bar{\boldsymbol{\theta}}_i$ almost everywhere with respect to the Lebesgue measure in $\mathcal{X}_i$.

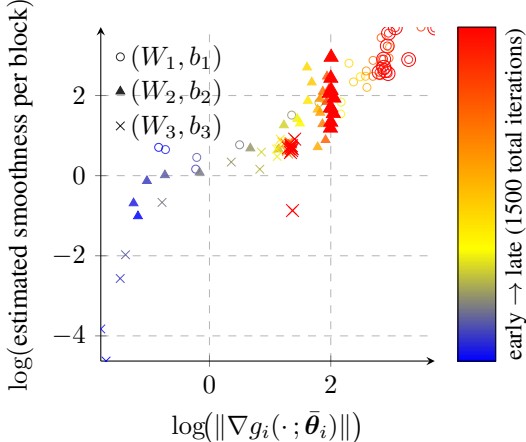

Figure 1: Estimated smoothness constant of $g_i(\cdot; \bar{\boldsymbol{\theta}}_i)$ in (3.2) vs its gradient norm. For more details of the experiment see Appendix A.6.1

It is possible to relate $\ell$-smooth to its first-order reminiscent, known as $(r, \ell)$-smoothness and vice-versa under specific choices for functions $r$ and $\ell$ (see Appendix A.4).

**Definition 2** ($(r, \ell)$-smoothness,(Li et al., 2024)). A real-valued differentiable function $g_i : \mathcal{X}_i \times \bar{\mathcal{X}}_i \to \mathbb{R}$ is $(r, \ell)$-smooth for continuous functions $r, \ell : [0, +\infty) \to (0, +\infty)$ where $\ell$ is non-decreasing and $r$ is non-increasing, if for any $\theta_i \in \mathcal{X}_i$ we have $\mathcal{B}(\theta_i, r(\|\nabla g_i(\theta_i; \bar{\boldsymbol{\theta}}_i)\|)) \subseteq \mathcal{X}_i$ and, for all $\theta_i^1, \theta_i^2 \in \mathcal{B}(\theta_i, r(\|\nabla g_i(\theta_i; \bar{\boldsymbol{\theta}}_i)\|))$ it holds that $\|\nabla g_i(\theta_i^1; \bar{\boldsymbol{\theta}}_i) - \nabla g_i(\theta_i^2; \bar{\boldsymbol{\theta}}_i)\| \leq \ell(\|\nabla g_i(\theta_i; \bar{\boldsymbol{\theta}}_i)\|)\|\theta_i^1 - \theta_i^2\|$.

The $(r, \ell)$-smoothness requires successive updates distance $\|\boldsymbol{\theta}^{k+1} - \boldsymbol{\theta}^k\|$ to be bounded. Although in algorithms like Gradient Descent (GD), this is satisfied through a bounded gradient norm condition and the sequential form of the algorithm, in BDCA such a link is nontrivial. To solve this, we exploit the non-uniqueness of the DC decomposition by adding and subtracting $\frac{\rho}{2}\|\theta_{i_k}\|^2$ to (2.1) on each block, yielding the proximal-type subproblems in (4.4) and ensuring bounded iterate differences. Under the assumptions below, we propose the convergence guarantee for Algorithm 1.

**Assumption 2.** For every $i \in \{n\}$, the functions $g_i$ is differentiable and closed within its open domain $\mathcal{X}_i \times \bar{\mathcal{X}}_i$.

**Assumption 3.** For every $i \in \{n\}$, the functions $h_i$ are Lipschitz continuous with constant R.

**Theorem 4.2.** *Consider Assumptions 2 and 3 when $\boldsymbol{\theta}^k$ is the output of Algorithm 1 for any initialization $\boldsymbol{\theta}^0 \in \mathcal{X}$. Then, for any $\ell$-smooth $g_i$ with subquadratic $\ell$, if $h_{i_k}(\theta_{i_k}^k; \bar{\boldsymbol{\theta}}_{i_k}^k) - h_{i_0}(\theta_{i_0}^0; \bar{\boldsymbol{\theta}}_{i_0}^0) \leq H$ for a constant $H \geq 0$, $E := \sup\{u > 0 : u^2 \leq 2\ell(2u).G\} < \infty$, $G := \max_j g_j(\theta_j^0; \bar{\boldsymbol{\theta}}_j^0) - g^* + H$ and $L := \ell(2E)$, then the sequence $\boldsymbol{\theta}^k$ generated by Algorithm 1 with $\rho \geq L\frac{2(E+R)}{E}$ will satisfy*

$$\min_{k \in \{1,\dots,K\}} \mathbb{E}_i\left[\mathcal{G}^2(\boldsymbol{\theta}^k)\right] \leq \frac{2n(L+\rho)}{K}\left(f(\boldsymbol{\theta}^1) - f^\star\right). \tag{4.3}$$

For a detailed discussion on the convergence result and the proof of Theorem 4.2 see Appendix A.2. Compared to (Li et al., 2024), this rate is scaled by $n$ which is expected due to the random choice of the blocks in each iteration of Algorithm 1.

## 4.3 STOCHASTIC PROXIMAL BDCA UNDER GENERALIZED SMOOTHNESS

In this section, we target (2.1) when on $i^{\text{th}}$ block we have

$$f(\boldsymbol{\theta}) := g_i(\theta_i; \bar{\boldsymbol{\theta}}_i) - h_i(\theta_i; \bar{\boldsymbol{\theta}}_i) = \mathbb{E}_{s \sim \mathbb{P}}[g_i(\theta_i; \bar{\boldsymbol{\theta}}_i, s) - h_i(\theta_i; \bar{\boldsymbol{\theta}}_i, s)] \tag{4.5}$$

---

**Algorithm 1** Proximal BDC

---

**Input:** set $k = 0$, and number of blocks $n$, number of iterations $T$
REPEAT:
Randomly choose $i_k$ in $[n]$ with uniform distribution
Evaluate $u_{i_k}^k \in \partial h_{i_k}(\theta_{i_k}^k; \bar{\boldsymbol{\theta}}_{i_k}^k)$,

$$\theta_{i_k}^{k+1} \in \underset{\boldsymbol{\theta}_{i_k} \in \mathcal{X}_{i_k}}{\operatorname{argmin}} \, g_{i_k}(\theta_{i_k}; \bar{\boldsymbol{\theta}}_{i_k}^k) - \langle u_{i_k}^k, \theta_{i_k} \rangle + \frac{\rho}{2} \|\theta_{i_k}^k - \theta_{i_k}\|^2 \tag{4.4}$$

Update $\boldsymbol{\theta}^{k+1} = \bar{\boldsymbol{\theta}}_{i_k}^k + \boldsymbol{\theta}_{i_k}^{k+1}$,
Set $k = k + 1$,
UNTIL Stopping criterion.

---

where $(\Omega, \Sigma_\Omega, \mathbb{P})$ is the probability space and BDC functions $g(., s), h(., s), s \in \Omega$ are defined on $\mathcal{X}$. In the realm of supervised learning, empirical loss is a common realistic approximation of the objective (4.5). In this sense, for each $i \in [n]$ we have

$$g_i(\theta_i; \bar{\boldsymbol{\theta}}_i) - h_i(\theta_i; \bar{\boldsymbol{\theta}}_i) \approx \frac{1}{J} \sum_{j=1}^{J} g_i(\theta_i; \bar{\boldsymbol{\theta}}_i, s^j) - \frac{1}{J} \sum_{j=1}^{J} h_i(\theta_i; \bar{\boldsymbol{\theta}}_i, s^j) \tag{4.6}$$

for $s^j \in \Omega$. For simplicity, denote $\hat{g}_i(\theta_i; \bar{\boldsymbol{\theta}}_i) = g_i(\theta_i; \bar{\boldsymbol{\theta}}_i, s)$ and $\hat{h}_i(\theta_i; \bar{\boldsymbol{\theta}}_i) = h_i(\theta_i; \bar{\boldsymbol{\theta}}_i, s)$. Throughout this section, we make the following assumption:

**Assumption 4.** Take $u_i \in \partial h_i(\theta_i; \bar{\boldsymbol{\theta}}_i)$ and $\hat{u}_i, \nabla \hat{g}_i(\theta_i; \bar{\boldsymbol{\theta}}_i)$ as the unbiased stochastic approximations of $u_i$ and $\nabla g_i(\theta_i; \bar{\boldsymbol{\theta}}_i)$ such that $\mathbb{E}[\hat{u}_i] = u_i$ and $\mathbb{E}[\nabla \hat{g}_i(\theta_i; \bar{\boldsymbol{\theta}}_i)] = \nabla g_i(\theta_i; \bar{\boldsymbol{\theta}}_i)$ with $\mathbb{E}[\|\nabla \hat{g}_i(\theta_i; \bar{\boldsymbol{\theta}}_i) - \hat{u}_i - (\nabla g_i(\theta_i; \bar{\boldsymbol{\theta}}_i) - u_i)\|^2] \leq \sigma^2$ for $i = 1, \ldots, n$.

To solve the stochastic minimization of (4.6), we need to modify Algorithm 1. Using i.i.d. random $s^k \sim \text{Unif}\{1, J\}$, we evaluate $\hat{u}_{i_k}^k \in \partial \hat{h}_{i_k}(\theta_{i_k}^k; \bar{\boldsymbol{\theta}}_{i_k}^k)$. Now, instead of (4.4), we solve:

$$\theta_{i_k}^{k+1} \in \underset{\boldsymbol{\theta}_{i_k} \in \mathcal{X}_{i_k}}{\operatorname{argmin}} \, g_{i_k}(\theta_{i_k}; \bar{\boldsymbol{\theta}}_{i_k}^k, s^k) - \langle \hat{u}_{i_k}^k, \theta_{i_k} \rangle + \frac{\rho}{2} \|\theta_{i_k}^k - \theta_{i_k}\|^2. \tag{4.7}$$

The following theorem formulates the convergence of SBDC algorithm explained above.

**Theorem 4.3.** *Consider assumptions 2, 3, and 4 when $\boldsymbol{\theta}^k$ as the output of (4.7) for any initialization $\boldsymbol{\theta}^0 \in \mathcal{X}$. Then, for any $\ell$-smooth $g_i$ with subquadratic $\ell$ take $g_{i_k}(\theta_{i_k}^k; \bar{\boldsymbol{\theta}}_{i_k}^k) - g^* \leq G$ and $\|\nabla \hat{g}_{i_k}(\theta_{i_k}^k; \bar{\boldsymbol{\theta}}_{i_k}^k) - \hat{u}_{i_k}^k - (\nabla g_{i_k}(\theta_{i_k}^k; \bar{\boldsymbol{\theta}}_{i_k}^k) - u_{i_k}^k)\| \leq F'$ for $G, F' > 0$ and $\rho \geq L \frac{2(E+R+F')}{E}$, $L := \ell(2E)$, $h_{i_k}(\theta_{i_k}^k; \bar{\boldsymbol{\theta}}_{i_k}^k) - h_{i_0}(\theta_{i_0}^0; \bar{\boldsymbol{\theta}}_{i_0}^0) \leq H$ for a constant $H \geq 0$. Further, for any $0 < \delta < 1$ consider $G := \max_j 8 \left(g_j(\theta_j^0; \bar{\boldsymbol{\theta}}_j^0) - g^* + C'\right)/\delta$, $C' := {}^{K\sigma^2}/\rho + H$, $F' = {}^{E\rho}/9L - (E + R)$, $\sigma^2 = \mathcal{O}(1/\sqrt{K})$, $\rho = \left(18L + \frac{9ER}{G} + \frac{81L}{4}\left[\frac{C'-H}{C'}\right]\right)\sqrt{K}$, $E := \sup\{u > 0 : u^2 \leq 2\ell(2u)G\} < \infty$, and $K \geq {}^{(L+\frac{3}{2}\rho)nG\delta}/4\epsilon^2$ for any $\epsilon > 0$. Then, with probability at least $1 - \delta$ the iterates of the (4.7) with $n$ blocks will satisfy*

$$\min_{k=1,\ldots,K} \mathbb{E}_{s,i}\left[\mathcal{G}^2(\boldsymbol{\theta}^k)\right] \leq \epsilon^2. \tag{4.8}$$

The proof of Theorem 4.3 with detailed discussion is presented in Appendix A.3. This result achieves the gradient complexity $\mathcal{O}(n^2/\epsilon^4)$ for $\rho = \Omega(\sqrt{K})$. The condition $\sigma^2 = \mathcal{O}(1/\sqrt{K})$ is achievable through a comparable number of samples in the mini-batch or through variance reduction techniques. In particular, by Lemma A.12 (see Appendix A.5) batch size should be $\Omega(n/\epsilon^2)$ and this means a sample complexity $\Omega(n^3/\epsilon^6)$. Similar assumption has appeared in previous works such as (Nitanda and Suzuki, 2017; Yurtsever et al., 2019).

## 5 APPLICATIONS

We highlight the versatility of the BDC framework through a few illustrative applications and numerical experiments.

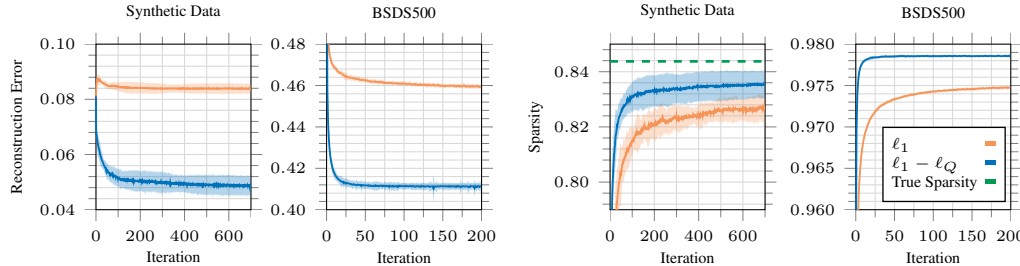

Figure 2: Reconstruction error and sparsity of codes for the $\ell_1$ regularizer (orange) and the nonconvex $\ell_1 - \ell_Q$ regularizer (blue) on synthetic data and BSDS500 patches. Solid curves denote the mean over 10 runs, with shaded bands showing 95% probabilistic bounds. The dashed green line indicates the true sparsity level in the synthetic data. The nonconvex $\ell_1 - \ell_Q$ formulation yields both lower reconstruction error and sparser codes.

**Proximal Alternating Linearized Minimization.** The general class of nonconvex nonsmooth optimization problems in the form of:

$$\min_{\theta_i \in \mathcal{X}_{d_i}, i=1,\ldots,n} \sum_{i=1}^{n} f_i(\theta_i) + H(\boldsymbol{\theta}), \tag{5.1}$$

was addressed by Bolte et al. (2014) for an $L$-smooth function $H(\boldsymbol{\theta})$. This problem is an instance of (2.1) under some assumptions. Bolte et al. (2014) proved a non-asymptotic convergence rate for the PALM algorithm assuming the KL property while in this work, we do not make such assumption.

**Multiplicative Multitask Feature Learning.** MMFL aims to train a neural network that learns shared representations across multiple tasks. A shared vector $\boldsymbol{c} \in \mathbb{R}^T$ indicates feature usefulness for $T$ tasks, and is multiplied by the weight vector $\beta_t \in \mathbb{R}^d$, where $d$ is the number of features. Sparse regularization is then to exclude redundant features. For details on regularizer choices, see (Wang et al., 2016). The mathematical formulation of the MMFL problem with sparse regularizer on $\boldsymbol{c}$ is:

$$\min_{\boldsymbol{c} \geq 0, \beta_t} \sum_{t=1}^{T} \text{loss}(\text{diag}(\boldsymbol{c})\beta_t, X_t, y_t) + \lambda_1 \sum_{t=1}^{T} \|\beta_t\|_p^p + \lambda_2 \|\boldsymbol{c}\|_0, \tag{5.2}$$

where $\text{loss}(\cdot)$ denotes a loss function (e.g., least squares or logistic loss), $X_t \in \mathbb{R}^{n_t \times d}$ is the dataset, and $y_t$ represents the labels for the $t^{\text{th}}$ task. Since convex $\ell_1$ regularizers are too relaxed to approximate the shrinkage effect in the feature space, non-convex alternatives such as $\|\boldsymbol{c}\|_1 - \|\boldsymbol{c}\|_Q$ ($\|\mathbf{x}\|_Q$ denotes the largest-$Q$ norm) or the capped $\ell_1$-norm ($\sum_t \min\{|\boldsymbol{c}_t|, \gamma\} = \|\boldsymbol{c}\|_1 - \sum_t \max\{|c_t| - \gamma, 0\}$) are preferred (Gong et al., 2012). Replacing either of these in (5.2) results in a BDC optimization task.

**Rank Regularization.** Consider an optimization problem of the following form:

$$\min_{X,Y} f(X, Y) + \lambda \, \text{rank}(X) \tag{5.3}$$

where $X$ and $Y$ are two matrices in $\mathbb{R}^{n \times m}$ and the function $f(\cdot)$ is BDC. This type of problem has several applications, such as matrix completion (Hazan et al., 2023) and deep learning (Wang et al., 2024; Scarvelis and Solomon, 2024). Due to the rank term, (5.3) is NP-hard and a convex surrogate known as the nuclear norm $\|X\|_* = \sum_{i=1}^{\min\{n,m\}} \sigma_i$ is often utilized, where $\sigma_i$ represents the $i$-th largest singular value. A tighter non-convex approximation of the rank regularizer is the truncated nuclear norm (TNN), defined as $\sum_{i=r+1}^{\min\{n,m\}} \sigma_i$. TNN can be rewritten as $\|X\|_* - \sum_{i=1}^{r} \sigma_i$, which is a DC function. Thus, replacing it in (5.3) gives a BDC due to the DC regularizer. Note that when $r = 1$, the regularizer is equivalent to $\|X\|_* - \|X\|_2$, which is a special case commonly used as a non-convex regularizer for the rank term (Jiang et al., 2021).

**Sparse Dictionary Learning.** We illustrate the applicability of our theoretical framework on SDL problem. Given a data matrix $Y = [y_1, \ldots, y_n] \in \mathbb{R}^{m \times n}$, SDL seeks a dictionary $\mathbf{D} = [d_1, \ldots, d_k] \in \mathbb{R}^{m \times k}$ and sparse codes $X = [x_1, \ldots, x_n] \in \mathbb{R}^{k \times n}$ by solving

$$\min_{\mathbf{D} \in \mathcal{C}, X} \sum_{i=1}^{n} \frac{1}{2} \|y_i - \mathbf{D} x_i\|_2^2 + \alpha \sum_{i=1}^{n} \|x_i\|_0, \quad \mathcal{C} = \{\mathbf{D} \in \mathbb{R}^{m \times k} \mid \|d_j\|_2 \leq 1 \, \forall j\}. \tag{5.4}$$

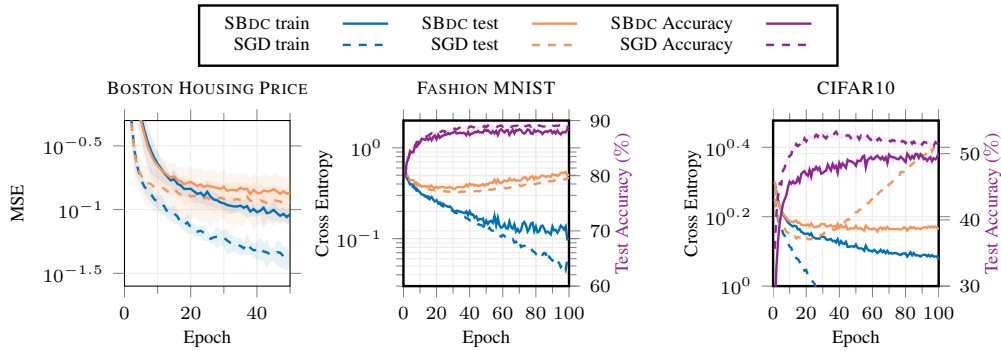

Figure 3: Comparison of SBDC with SGD in regression (left) and classification (middle, right) for 10 Monte-Carlo instances. The shaded bands specify the 68% confidence intervals. As depicted, SBDC has comparable performance to the SGD in terms of the test loss and test accuracy.

Since the $\ell_0$-norm is NP-hard to optimize, it is often replaced with $\ell_1$-norm. More recently, nonconvex regularizers have been used to yield a tighter approximation to sparsity. Following (Deng and Lan, 2020; Maskan et al., 2024), we consider

$$\min_{\mathbf{D}\in\mathcal{C},\,X}\sum_{i=1}^{n}\frac{1}{2}\|y_i - \mathbf{D}\,x_i\|_2^2 + \alpha\sum_{i=1}^{n}\big(\|x_i\|_1 - \|x_i\|_Q\big). \tag{5.5}$$

Problem (5.5) is BDC: fixing either $\mathbf{D}$ or $X$ yields a DC problem. The optimization problem (5.5) is a special case of our formulation in Section 4.1 and Appendix A.1. We conducted numerical simulations to solve the SDL problem with $\ell_1$ and nonconvex $\ell_1 - \ell_Q$ regularizers (Eq. 5.5) via BDCA (4.1). Performance is measured by reconstruction error $\|Y - \mathbf{D}X\|_F^2$ and the proportion of zeros in $X$. We compare using synthetic data and Berkeley segmentation dataset (Martin et al., 2001). The results are shown in Figure 2. For more detail, see Appendix A.6.

**Application to Neural Networks.** In Section 3.2 we found explicit formulations of training objective for MSE and CE losses as a BDC problem. Using the these formulations and (4.7), we train neural networks for the MSE and the CE loss functions. Next, we train neural networks using (4.7). We use CIFAR10 and FASHIONMNIST datasets for the classification task and BOSTON HOUSING PRICE dataset[1] for the regression task. See Appendix A.6 for a details of our implementation setting.

**Remark 5.1.** *Our implementation via (4.7) computes gradients only with respect to the selected random layer in each iteration, offering computational benefits by reducing the gradient calculation bottleneck. In practice, we backpropagate only up to the selected layer.*

## 6 CONCLUSION AND DISCUSSION

We introduce and motivate the *multi-block* DC (BDC) class—strictly richer than classical DC—and demonstrate its practicality from two angles: (i) compared to DC decompositions, BDC formulations are far cheaper to construct (e.g., exponentially cheaper for monomials), and (ii) obtaining BDC decompositions for modern problems (e.g., training deep ReLU networks) is vastly easier and constructive. Subsequently, after developing foundational properties of the BDC class, we leverage multi-block convexity to propose a Gauss–Seidel–type BDC algorithm with non-asymptotic guarantees under $L$-smoothness, generalized smoothness, and stochasticity. Applications to MMFL, rank regularization, sparse dictionary learning, and neural network training illustrate the framework's practicality and breadth.

We conclude by noting one avenue for future work and two algorithmic limitations. On the theory side, a natural direction is to further investigate the representation complexity gap between the BDC and DC classes (e.g., for ReLU networks). On the algorithmic side, although Algorithm 1 ensures monotone descent, our analysis assumes bounded $g_i(\theta_i; \bar{\theta}_i)$ along the trajectory, which we enforce via bounded $h_i(\theta_i; \bar{\theta}_i)$ at update points; removing this assumption would strengthen the result. In addition, our generalized-smoothness theory currently covers only unconstrained BDC optimization; extending it to constrained problems remains open.

---

[1] https://www.kaggle.com/code/prasadperera/the-boston-housing-dataset

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

---

**Algorithm 2** BDC Algorithm (L-smooth))

---

**Input:** set $k = 0$, and number of blocks $n$, number of iterations $T$
REPEAT:
Randomly choose $i_k$ in $[1, ..., n]$ with uniform distribution
Evaluate $u_{i_k}^k \in \partial h_{i_k}(\theta_{i_k}^k; \bar{\boldsymbol{\theta}}_{i_k}^k)$,
Solve

$$\theta_{i_k}^{k+1} \in \underset{\theta_{i_k} \in \mathcal{M}^{i_k}}{\operatorname{argmin}}\ g_{i_k}(\theta_{i_k}^k; \bar{\boldsymbol{\theta}}_{i_k}^k) + r_{i_k}(\theta_{i_k}) - \langle u_{i_k}^k, \theta_{i_k} \rangle \tag{A.4}$$

Update $\boldsymbol{\theta}^{k+1} = \bar{\boldsymbol{\theta}}_{i_k}^k + \boldsymbol{\theta}_{i_k}^{k+1}$,
Set $k = k + 1$,
UNTIL Stopping criterion.

---

## A DISCUSSIONS

In this section, we provide more general results under smoothness assumption in Appendix A.1, background on generalized smoothness in Appendix A.4, useful lemmas in stochastic gradient estimator's variance in Appendix A.5, and more detail on numerical results in Appendix A.6. All the proofs are given in Appendix B.

### A.1 MULTI-BLOCK DCA UNDER SMOOTHNESS ASSUMPTION

Here, we focus on a more general problem of the form:

$$\min_{\boldsymbol{\theta} \in \mathcal{M}}\ f(\boldsymbol{\theta}), \tag{A.1}$$

where for each block $\boldsymbol{\theta}_i$

$$f(\boldsymbol{\theta}) := g_i(\theta_i; \bar{\boldsymbol{\theta}}_i) + r_i(\theta_i) - h_i(\theta_i; \bar{\boldsymbol{\theta}}_i), \tag{A.2}$$

when $g_i(\cdot; \bar{\boldsymbol{\theta}}_i)$ is an $L$-smooth function, $r_i(\theta_i)$ is a non-differentiable convex function, and we have constraint set $\mathcal{M} = \mathcal{M}_1 \times \mathcal{M}_2 \times \ldots \times \mathcal{M}_n$ and each $\mathcal{M}_i \subseteq \mathbb{R}^{d_i}$ is a closed convex set and $d = \sum_{i=1}^n d_i$. The rest of the setup is similar to the unconstrained setting in Section 2. Problem (A.1) was addressed for DC objective function $f(\boldsymbol{\theta})$ in Maskan et al. (2024). Here, we show that our formulation is capable of solving such problem formulation under a multi-block DC assumption on the objective $f(\boldsymbol{\theta})$. Under this assumption, we propose a multi-block DCA (BDCA) algorithm, shown in Algorithm 2. The following Theorem shows the convergence of this method. The proof of this theorem is given in Appendix B.11.

**Theorem A.1.** *The sequence generated by the update* (A.4) *will satisfy*

$$\min_{k \in \{1, ..., K\}} \mathbb{E}_i \left[ \operatorname{gap}_{\mathcal{M}}^L(\boldsymbol{\theta}^k) \right] \le \frac{n}{K} \left( f(\boldsymbol{\theta}^1) - f^\star \right), \tag{A.3}$$

*where $\mathbb{E}_i[.]$ denotes expectation w.r.t. the $i^{th}$ block choice and*

$$\operatorname{gap}_{\mathcal{M}}^L(\boldsymbol{y}) := \max_{\boldsymbol{x} \in \mathcal{M}} \min_{\boldsymbol{\nu} \in \partial f(\boldsymbol{y})} \left\{ \langle \boldsymbol{\nu}, \boldsymbol{y} - \boldsymbol{x} \rangle + r(\boldsymbol{y}) - r(\boldsymbol{x}) - \frac{L}{2} \|\boldsymbol{x} - \boldsymbol{y}\|^2 \right\},$$

*is a gap measure ensuring convergence to first order stationary points and we denote $\partial f(\boldsymbol{y}) = \nabla g_i(y_i; \bar{\boldsymbol{y}}_i) - u_i$ for $u \in \partial h_i(y_i; \bar{\boldsymbol{y}}_i)$.*

This result, is more general than the one presented in Corollary 4.1. Specifically, when $r$ doesn't exist and $\mathcal{M}$ becomes the domain $\mathcal{X}$ (no constraint), the gap measure is $\frac{1}{2L} \mathcal{G}(\boldsymbol{\theta}^k)$ for $\mathcal{G}(\boldsymbol{\theta}) := \sup_{u \in \partial f(\boldsymbol{\theta})} \|u\|$ which results in Corollary 4.1.

### A.2 DETAILED ANALYSIS OF MULTI-BLOCK PROXIMAL DCA UNDER GENERALIZED SMOOTHNESS ASSUMPTION

In this section, we provide a more detailed discussion and prove the results in Section 4.2. Recall that we assumed a more relaxed assumption on the component $g_i(\cdot; \bar{\boldsymbol{\theta}}_i)$, known as $\ell$-smoothness. A first order reminiscent of the $\ell$-smoothness is the $(r, \ell)$-smoothness.

**Definition 3** ($(r, \ell)$-smoothness,(Li et al., 2024))**.** A real-valued differentiable function $g_i : \mathcal{X}_i \times \bar{\mathcal{X}}_i \to \mathbb{R}$ is $(r, \ell)$-smooth for continuous functions $r, \ell : [0, +\infty) \to (0, +\infty)$ where $\ell$ is non-decreasing and $r$ is non-increasing, if for any $\theta_i \in \mathcal{X}_i$ we have $\mathcal{B}(\theta_i, r(\|\nabla g_i(\theta_i; \bar{\boldsymbol{\theta}}_i)\|)) \subseteq \mathcal{X}_i$ and, for all $\theta_i^1, \theta_i^2 \in \mathcal{B}(\theta_i, r(\|\nabla g_i(\theta_i; \bar{\boldsymbol{\theta}}_i)\|))$ it holds that $\|\nabla g_i(\theta_i^1; \bar{\boldsymbol{\theta}}_i) - \nabla g_i(\theta_i^2; \bar{\boldsymbol{\theta}}_i)\| \leq \ell(\|\nabla g_i(\theta_i; \bar{\boldsymbol{\theta}}_i)\|)\|\theta_i^1 - \theta_i^2\|$.

Due to $\|\nabla g_i(\theta_i; \bar{\boldsymbol{\theta}}_i)\| \leq \|\nabla g_j(\theta_j, \bar{\boldsymbol{\theta}}_j)\|$ for $j := \arg\max_k \|\nabla g_k(\theta_k, \bar{\boldsymbol{\theta}}_k)\|$ and the fact that $r$ is a non-increasing function, we get $\mathcal{B}(\theta_i, r(\|\nabla g_j(\theta_j, \bar{\boldsymbol{\theta}}_j)\|)) \subseteq \mathcal{B}(\theta_i, r(\|\nabla g_i(\theta_i; \bar{\boldsymbol{\theta}}_i)\|))$. Therefore, for any $\theta_i^1, \theta_i^2 \in \mathcal{B}(\theta_i, r(\|\nabla g_j(\theta_j, \bar{\boldsymbol{\theta}}_j)\|))$ that satisfy $(r, \ell)$-smoothness, we have:

$$\|\nabla g_i(\theta_i^1; \bar{\boldsymbol{\theta}}_i) - \nabla g_i(\theta_i^2; \bar{\boldsymbol{\theta}}_i)\| \leq \ell(\|\nabla g_j(\theta_j, \bar{\boldsymbol{\theta}}_j)\|)\|\theta_i^1 - \theta_i^2\|.$$

It is possible to relate these two definitions, i.e., we can show that an $\ell$-smooth function is $(r, \ell)$-smooth and vice-versa under specific choices for $r$ and $\ell$. This connection, investigated by Li et al. (2024), with more discussion and related results are given in Appendix A.4.

A necessary condition for $(r, \ell)$-smoothness is that the iterates of our sequential algorithm have a bounded distance $\|\boldsymbol{\theta}^{k+1} - \boldsymbol{\theta}^k\|$. Usually, this is satisfied through bounded gradient norm condition and the sequential form of the algorithm. For example, in GD we have $\|\boldsymbol{\theta}^{k+1} - \boldsymbol{\theta}^k\| = \|\eta\nabla f(\boldsymbol{\theta}^k)\|$. In DCA, such a connection does not have trivial validity. Using the non-uniqueness of the DC decomposition, we add and subtract $\frac{\rho}{2}\|\theta_{i_k}\|^2$ to (2.1) on each block. This gives the subproblems (4.4) after applying DCA, which are proximal-type updates. The expected convergence rate of Algorithm 1 is finalized in the following proposition:

**Proposition A.2.** *Consider Assumptions 2 and 3 when $\boldsymbol{\theta}^k$ is the output of Algorithm 1 for any initialization $\boldsymbol{\theta}^0 \in \mathcal{X}$. Then, for any $\ell$-smooth $g_i$ with subquadratic $\ell$, if $h_{i_k}(\theta_{i_k}^k; \bar{\boldsymbol{\theta}}_{i_k}^k) - h_{i_0}(\theta_{i_0}^0; \bar{\boldsymbol{\theta}}_{i_0}^0) \leq H$ for a constant $H \geq 0$, $E := \sup\{u > 0 : u^2 \leq 2\ell(2u).G\} < \infty$, $G := \max_j g_j(\theta_j^0; \bar{\boldsymbol{\theta}}_j^0) - g^* + H$ and $L := \ell(2E)$, then the sequence $\boldsymbol{\theta}^k$ generated by Algorithm 1 with $\rho \geq L\frac{2(E+R)}{E}$ will satisfy*

$$\min_{k \in \{1, \ldots, K\}} \mathbb{E}_i\left[\mathcal{G}^2(\boldsymbol{\theta}^k)\right] \leq \frac{2n(L + \rho)}{K}\left(f(\boldsymbol{\theta}^1) - f^\star\right). \tag{A.5}$$

*Proof.* We begin by bounding the updates through the following lemma. See Appendix B.5 for the proof.

**Lemma A.3.** *For any starting point $\boldsymbol{\theta}^k$ the update generated by (4.4) is in $\mathcal{B}\left(\boldsymbol{\theta}^k, \frac{2}{\rho}\mathcal{G}(\boldsymbol{\theta}^k)\right)$.*

This result guarantees $\|\boldsymbol{\theta}^{k+1} - \boldsymbol{\theta}^k\| \leq \frac{2}{\rho}\sup_{\nu_{i_k}^k \in \partial_{i_k} f(\boldsymbol{\theta}^k)} \|\nu_{i_k}^k\| \leq \frac{2}{\rho}\mathcal{G}(\boldsymbol{\theta}^k)$. Due to $\|\nu_{i_k}^k\| \leq \|\nabla g_{i_k}(\theta_{i_k}^k; \bar{\boldsymbol{\theta}}_{i_k}^k)\| + \|u_{i_k}^k\|$ for any $u_{i_k}^k \in \partial h_{i_k}(\theta_{i_k}^k; \bar{\boldsymbol{\theta}}_{i_k}^k), \nu_{i_k}^k \in \partial_{i_k} f(\boldsymbol{\theta}^k)$, and $R$-Lipschitz $h_{i_k}$ (see Assumption 3), we need to bound $\|\nabla g_{i_k}(\theta_{i_k}^k; \bar{\boldsymbol{\theta}}_{i_k}^k)\|$ in order to have a bounded $\|\nu_{i_k}^k\|$. When $g_{i_k}$ is $\ell$-smooth with bounded $g_{i_k}(\theta_{i_k}; \bar{\boldsymbol{\theta}}_{i_k}^k) - g^*$ for some $\theta_{i_k} \in \mathcal{X}_{i_k}$, we get $\|\nabla g_{i_k}(\theta_{i_k}; \bar{\boldsymbol{\theta}}_{i_k}^k)\| \leq E$ for $E > 0$ (see Corollary A.11 in Appendix A.4). The following Lemma bounds $g_{i_k}(\theta_{i_k}^k; \bar{\boldsymbol{\theta}}_{i_k}^k) - g^*$ and proposes a choice for $\rho$ such that we have local bound on the gradients.

**Lemma A.4.** *Consider Assumptions 2 and 3 when $\boldsymbol{\theta}^{k+1}$ is the output of Algorithm 1 for any initialization $\boldsymbol{\theta}^0 \in \mathcal{X}$. Then, if $h_{i_k}(\theta_{i_k}^k; \bar{\boldsymbol{\theta}}_{i_k}^k) - h_{i_0}(\theta_{i_0}^0; \bar{\boldsymbol{\theta}}_{i_0}^0) \leq H$ for a constant $H \geq 0$, we have $g_{i_k}(\theta_{i_k}^k; \bar{\boldsymbol{\theta}}_{i_k}^k) - g^* \leq g_{i_0}(\theta_{i_0}^0; \bar{\boldsymbol{\theta}}_{i_0}^0) - g^* + H$. Additionally, for any $\ell$-smooth $g_i$ with subquadratic $\ell$, if $\rho \geq \ell(2E)\frac{2(E+R)}{E}$, then for any $i \in [n]$ and $\boldsymbol{\theta}^1, \boldsymbol{\theta}^2 \in \mathcal{B}(\boldsymbol{\theta}^k, 2(E+R)/\rho)$ we have:*

$$\begin{aligned}
\|\nabla g_i(\theta_i^2; \bar{\boldsymbol{\theta}}_i^2) - \nabla g_i(\theta_i^1; \bar{\boldsymbol{\theta}}_i^1)\| &\leq L\|\theta_i^1 - \theta_i^2\|, \\
g_i(\theta_i^2; \bar{\boldsymbol{\theta}}_i^2) &\leq g_i(\theta_i^1; \bar{\boldsymbol{\theta}}_i^1) + \langle\nabla g_i(\theta_i^1; \bar{\boldsymbol{\theta}}_i^1), \theta_i^2 - \theta_i^1\rangle + \frac{L}{2}\|\theta_i^1 - \theta_i^2\|^2,
\end{aligned} \tag{A.6}$$

*where $L = \ell(2E)$ is the effective smoothness for some $E > 0$.*

See Appendix B.6 for the proof. Note that $\boldsymbol{\theta}^1, \boldsymbol{\theta}^2$ in Lemma A.4 differ only in their $i_k^{\text{th}}$ block selected on iteration $k$ of Algorithm 1. Now building on the previous lemmas, we propose our main convergence result for Algorithm 1. The proof of this result is given in Appendix B.7

**Proposition A.5.** *Assume the conditions in Lemma A.4 and take $E := \sup\{u > 0 : u^2 \leq 2\ell(2u).G\} < \infty$, $G := \max_i g_i(\theta_i^0; \bar{\boldsymbol{\theta}}_i^0) - g^* + H$ and $L := \ell(2E)$. Then, the sequence $\boldsymbol{\theta}^k$ generated by Algorithm 1 with $\rho \geq L\frac{2(E+R)}{E}$ will satisfy*

$$\min_{k \in \{1,\ldots,K\}} \mathbb{E}_i\left[\mathcal{G}^2(\boldsymbol{\theta}^k)\right] \leq \frac{2n(L+\rho)}{K}\left(f(\boldsymbol{\theta}^1) - f^\star\right). \tag{A.7}$$

$\square$

## A.3 DETAILED ANALYSIS OF STOCHASTIC MULTI-BLOCK PROXIMAL DCA UNDER GENERALIZED SMOOTHNESS ASSUMPTION

In order to show the convergence of (4.7), we start by ensuring the boundedness of the updates as in the following lemma. See Appendix B.9 for the proof.

**Lemma A.6.** *Denote the sequence generated by (4.7) as $\boldsymbol{\theta}^k$. Then, for any $u_{i_k}^k \in \partial h_{i_k}(\theta_{i_k}^k; \bar{\boldsymbol{\theta}}_{i_k}^k)$, if $\nabla \hat{g}_{i_k}(\theta_{i_k}^k; \bar{\boldsymbol{\theta}}_{i_k}^k), \hat{u}_{i_k}^k$ are the respective stochastic approximations of $u_{i_k}^k$ and $\nabla g_{i_k}(\theta_{i_k}^k; \bar{\boldsymbol{\theta}}_{i_k}^k)$, we have:*

$$\|\boldsymbol{\theta}^{k+1} - \boldsymbol{\theta}^k\| \leq \frac{2}{\rho}\left(\mathcal{G}(\boldsymbol{\theta}^k) + \|\nabla \hat{g}_{i_k}(\boldsymbol{\theta}^k) - \hat{u}_{i_k}^k - (\nabla g_{i_k}(\theta_{i_k}^k; \bar{\boldsymbol{\theta}}_{i_k}^k) - u_{i_k}^k)\|\right).$$

Note that the bound in Lemma A.6 does not immediately imply that the solutions to the subproblems (4.7) will fall inside a ball. For this, we take $\|\nabla \hat{g}_{i_k}(\boldsymbol{\theta}^k) - \hat{u}_{i_k}^k - (\nabla g_{i_k}(\theta_{i_k}^k; \bar{\boldsymbol{\theta}}_{i_k}^k) - u_{i_k}^k)\| \leq F'$ for some $F' > 0$. Later, we find the value of $F'$ such that the bound $\|\nabla \hat{g}_{i_k}(\boldsymbol{\theta}^k) - \hat{u}_{i_k}^k - (\nabla g_{i_k}(\theta_{i_k}^k; \bar{\boldsymbol{\theta}}_{i_k}^k) - u_{i_k}^k)\| \leq F'$ holds with high probability. Then, a similar result to Lemma A.4 holds in the stochastic setting.

**Lemma A.7.** *Consider Assumptions 2 and 3 when $\boldsymbol{\theta}^k$ is the output of (4.7) for any initialization $\boldsymbol{\theta}^0 \in \mathcal{X}$. Then, for any $\ell$-smooth $g_i$ with subquadratic $\ell$ if $g_{i_k}(\theta_{i_k}^k; \bar{\boldsymbol{\theta}}_{i_k}^k) - g^* \leq G$ and $\|\nabla \hat{g}_{i_k}(\theta_{i_k}^k; \bar{\boldsymbol{\theta}}_{i_k}^k) - \hat{u}_{i_k}^k - (\nabla g_{i_k}(\theta_{i_k}^k; \bar{\boldsymbol{\theta}}_{i_k}^k) - u_{i_k}^k)\| \leq F'$ for $G, F' > 0$ and $\rho \geq L\frac{2(E+R+F')}{E}$ for $L := \ell(2E)$, we have:*

$$\|\nabla g_i(\theta_i^2; \bar{\boldsymbol{\theta}}_i^2) - \nabla g_i(\theta_i^1; \bar{\boldsymbol{\theta}}_i^1)\| \leq L\|\theta_i^1 - \theta_i^2\|,$$
$$g_i(\theta_i^2; \bar{\boldsymbol{\theta}}_i^2) \leq g_i(\theta_i^1; \bar{\boldsymbol{\theta}}_i^1) + \langle\nabla g_i(\theta_i^1; \bar{\boldsymbol{\theta}}_i^1), \theta_i^2 - \theta_i^1\rangle + \frac{L}{2}\|\theta_i^1 - \theta_i^2\|^2, \tag{A.8}$$

*for any $\boldsymbol{\theta}^1, \boldsymbol{\theta}^2 \in \mathcal{B}(\boldsymbol{\theta}^k, 2(E+R+F')/\rho)$.*

See Appendix B.10 for the proof. Note that if $F' = 0$, we get $\rho \geq 2L(E+R)/E$ which was in Lemma A.4. In order to use Lemma A.7, we need to show $g_{i_k}(\theta_{i_k}^k; \bar{\boldsymbol{\theta}}_{i_k}^k) - g^* \leq G$ and $\|\nabla \hat{g}_{i_k}(\theta_{i_k}^k; \bar{\boldsymbol{\theta}}_{i_k}^k) - \hat{u}_{i_k}^k - (\nabla g_{i_k}(\theta_{i_k}^k; \bar{\boldsymbol{\theta}}_{i_k}^k) - u_{i_k}^k)\| \leq F'$. Due to stochasticity, it is not possible to directly bound these values for all the iterations. Instead, we will show that the probabilities of the following events are low up to time $K$:

$$t_1 := \min\left\{\min\{k|g_{i_k}(\theta_{i_k}^{k+1}; \bar{\boldsymbol{\theta}}_{i_k}^k) - g^* > G\}, K\right\},$$
$$t_2 := \min\left\{\min\{k|\|\nabla \hat{g}_{i_k}(\theta_{i_k}^k; \bar{\boldsymbol{\theta}}_{i_k}^k) - \hat{u}_{i_k}^k - (\nabla g_{i_k}(\theta_{i_k}^k; \bar{\boldsymbol{\theta}}_{i_k}^k) - u_{i_k}^k)\| > F'\}, K\right\}, \tag{A.9}$$
$$t := \min\{t_1, t_2\},$$

In (A.9), the event $t_1 = K$ will ensure $g_{i_k}(\theta_{i_k}^k; \bar{\boldsymbol{\theta}}_{i_k}^k) - g^* \leq G$ before time $k < K$ and the event $t_2 = K$ will ensure $\|\nabla \hat{g}_{i_k}(\theta_{i_k}^k; \bar{\boldsymbol{\theta}}_{i_k}^k) - \hat{u}_{i_k}^k - (\nabla g_{i_k}(\theta_{i_k}^k; \bar{\boldsymbol{\theta}}_{i_k}^k) - u_{i_k}^k)\| \leq F'$ before time $k < K$. Next, we should show that the probability of the event $\{t < K\}$ is low. Alternatively, we can show a low probability for the event $\{t = t_2 < K\} \cup \{t = t_1 < K, t_2 = K\}$. This is a similar technique to (Li et al., 2024) in order to show convergence in the stochastic setting. Compared to their work, our proposed method in (4.7) targets a more general class of functions (BDC). Although the generality of our function class, our guarantee in Theorem 4.3 requires only the first components of our BDC structure to be $\ell$-smoothness. In this sense, our work generalizes the prior result by Li et al. (2024). The main convergence result is given in the following proposition (see Appendix B.8 for the proof).

**Proposition A.8.** *Consider Assumption 4 and the conditions in Lemma A.7 with $h_{i_k}(\theta_{i_k}^k; \bar{\boldsymbol{\theta}}_{i_k}^k) - h_{i_0}(\theta_{i_0}^0; \bar{\boldsymbol{\theta}}_{i_0}^0) \leq H$ for a constant $H \geq 0$. Further, for any $0 < \delta < 1$ take $G :=*

$\max_j 8 \left( g_j(\theta_j^0; \bar{\theta}_j^0) - g^* + C' \right) / \delta$, $C' := {}^{K\sigma^2}/\rho + H$, $F' = {}^{E\rho}/9L - (E + R)$, $\sigma^2 = \mathcal{O}(1/\sqrt{K})$, $\rho = (18L + \frac{9ER}{G} + \frac{81L}{4}\left[\frac{C'-H}{C'}\right])\sqrt{K}$, $E := \sup\{u > 0 : u^2 \leq 2\ell(2u)G\} < \infty$, $L := \ell(2E)$, *and* $K \geq {}^{(L+\frac{3}{2}\rho)nG\delta}/4\epsilon^2$ *for any* $\epsilon > 0$. *Then, with probability at least* $1 - \delta$ *the iterates of the* (4.7) *with* $n$ *blocks will satisfy*

$$\min_{k=1,\ldots,K} \mathbb{E}_{s,i}\left[\mathcal{G}^2(\theta^k)\right] \leq \epsilon^2. \tag{A.10}$$

## A.4 Background on $(r, \ell)$-smoothness and $\ell$-smoothness

Here, we discuss the required background and results on $\ell$-smoothness. We mainly represent the results from (Li et al., 2024) and briefly explain the results and connections with this work.

We start with the following lemma characterizing a local descent condition for any $x \in \mathcal{X}$ when $g$ is $(r, \ell)$-smooth:

**Lemma A.9** (Li et al. (2024)). *If* $g$ *is* $(r, \ell)$-*smooth, for any* $x \in \mathcal{X}$ *satisfying* $\|\nabla g(x)\| \leq E$ *we have* $\mathcal{B}(x, r(E)) \subset \mathcal{X}$, *and for any* $x_1, x_2 \in \mathcal{B}(x, r(E))$,

$$\|\nabla g(x_2) - \nabla g(x_1)\| \leq L\|x_1 - x_2\| \quad g(x_2) \leq g(x_1) + \langle \nabla g(x_1), x_2 - x_1 \rangle + \frac{L}{2}\|x_1 - x_2\|^2$$

*where* $L = \ell(E)$ *is the effective smoothness.*

The following proposition, bridges $\ell$-smoothness and $(r, \ell)$-smoothness. The importance of this result is due to the fact that it shows applicability of the descent Lemma A.9 on $\ell$-smooth functions.

**Proposition A.10** (Li et al. (2024)). *An* $(r, l)$-*smooth function is* $l$-*smooth; and an* $l$-*smooth is* $(r, m)$-*smooth with* $m(u) := l(u + a)$ *and* $r(u) := a/m(u)$ *for any* $a > 0$ *if* $f$ *is a closed function within its open domain* $\mathcal{X}$.

With this result, one can use Lemma A.9 on an $\ell$-smooth function which satisfies the conditions in Lemma A.9: bounded gradients and $(r, \ell)$-smoothness. Also, we need to ensure that the updates remain inside a ball. Despite the convexity of the function $g$ in our problem setup, DCA updates do not guarantee the boundedness of its gradients. Therefore, we use the following corollary which provides such bound when the function $\ell$ is sub-quadratic in the sense that $\lim_{t\to\infty} \ell(t)/t^2 = 0$.

**Corollary A.11** (Li et al. (2024)). *Suppose* $g$ *is* $\ell$-*smooth with sub-quadratic* $\ell$. *If* $g(x) - \inf_{y\in\mathcal{X}} g(y) \leq G$ *for some* $x \in \mathcal{X}$ *and* $G \geq 0$, *then* $E^2 = 2\ell(2E)G$ *and* $\|\nabla g(x)\| \leq E < \infty$ *for* $E := \sup\{u \geq 0 | u^2 \leq 2\ell(2u)G\}$

With Corollary A.11, if we can show that the updates remain inside a ball, then the descent condition in Lemma A.9 holds.

## A.5 Useful Lemma on Gradient Estimation Variance

The following lemma is a classical result on the variance in terms of the mini-batch size. We used this Lemma for the discussions on our reduced variance assumption in Theorem 4.3.

**Lemma A.12** (Lemma 2 from (Reddi et al., 2016) ). *Suppose that* $\mathcal{S}^k$ *is a subset that samples* $s^k$ *i.i.d realizations from the distribution* $\mathcal{P}$. *Let the stochastic estimator* $\nabla f(\theta^k, s^k)$ *satisfy the bounded variance condition* Assumption 4. *Then, the following bound holds:*

$$\mathbb{E}\left[\|\nabla f(\theta^k, s^k) - \nabla f(\theta^k)\|^2\right] \leq \frac{\sigma^2}{s^k}, \quad ,\forall\theta \in \mathcal{X}. \tag{A.11}$$

## A.6 More detail on Numerical Examples

In this section, we provide the reader with more detail of our implementation settings and parameter choices.

### A.6.1 GENERALIZED SMOOTHNESS ON DEEP NETWORKS.

The relationship between the Hessian of the objective function in training language models and the norm of its gradient was already observed in (Zhang et al., 2019). This relationship was later extended to more general cases by Li et al. (2024). The previous analyses, heavily relied on the trajectory of the optimization guided by GD updates. Here, we want to show that a similar relationship exists between the estimated smoothness of the first BDC component and its gradient norm when the updates are done by the BDCA. In order to do this, we use the same smoothness estimator as in (Santurkar et al., 2018) defined below:

$$\hat{L}_{g_i}(\boldsymbol{\theta}^k) = \max_{\gamma \in \{\delta, 2\delta, \ldots, 1\}} \frac{\nabla g_i(\theta_i^k + \gamma d) - \nabla g_i(\theta_i^k)}{\gamma d}, \tag{A.12}$$

for a small value $\delta$ and $d = \theta_i^{k+1} - \theta_i^k$. This value determines the variations along $d$ on the block $i$. Note that unlike previous results, in BDCA we do not necessarily decrease the value of the first component along the update trajectory. To show this, we conducted numerical simulations on a regression task using a three-layer ReLU network of size $(8 \times 64 \times 32 \times 1)$ on the California Housing dataset (Kelley Pace and Barry, 1997). We considered training for 30 epochs, a learning rate of $0.5 \times 10^{-3}$ with 10 oracle calls to the BDCA sub-problem solver. We set $\delta = 0.25$. The logarithm of the estimated smoothness constant of the first BDC component in (3.2) was depicted against its gradient norm for each block is depicted in Figure 1. This figure suggests that a sub-quadratic relationship between the layer-wise smoothness constant and their gradient norms exists, a similar relationship required for our convergence result in Theorem 4.2 and Theorem 4.3.

### A.6.2 SPARSE DICTIONARY LEARNING

Here, we explain the implementation structures of the sparse dictionary learning problem with more detail. Note that the structure of SDL problem fits with the more general analysis provided in Appendix A.1.

**Implementation Details.** Both formulations (with $\ell_1$ norm and (5.5)) are solved via alternating minimization. In each iteration, we first update $X$: for the $\ell_1$ model, we use GD; for the nonconvex model (5.5), we employ the DC algorithm by linearizing the $\|\cdot\|_Q$ term and then applying GD to the resulting convex surrogate. Next, we update $\mathbf{D}$ using a Frank–Wolfe procedure, projecting onto $\mathcal{C}$ to enforce the unit-$\ell_2$ constraints. A line search determines the optimal step size in each Frank–Wolfe update. We evaluate performance by the reconstruction error $\|Y - \mathbf{D}X\|_F^2$ and the proportion of zeros in $X$. Each experiment is repeated 10 times, and we report a 95% probabilistic bound in our plots. We compare the formulations on synthetic data and Berkeley segmentation dataset Martin et al. (2001).

**Synthetic Data.** We set $m = 10$, $k = 32$, and $n = 100$. A ground-truth dictionary $\mathbf{D}^* \in \mathbb{R}^{m \times k}$ is generated by sampling each entry i.i.d. from $\mathcal{N}(0, 1)$ and normalizing each column to unit $\ell_2$-norm. The true sparse code matrix $X^* \in \mathbb{R}^{k \times n}$ has exactly five nonzero entries per column, drawn i.i.d. from $\mathcal{N}(0, 1)$. We synthesize the data as $Y = \mathbf{D}^* X^*$, using $\alpha = 0.1$ and $Q = 5$. Results are shown in Figure 2.

In addition to the previous experiment, we also compare the loss behavior of the SDL problem under the GD and BDC algorithms for the nonconvex $\ell_1 - \ell_Q$ regularizer (Eq. 5.5). At each iteration, the gradient descent algorithm performs a single joint full-batch update of both the dictionary $D$ and the code matrix $X$, using the adaptive step size

$$\eta = \frac{1}{\|D\|_2^2 + \|X\|_2^2}$$

which is motivated by the block Lipschitz constants of the smooth reconstruction term. Dictionary feasibility is maintained by projecting each column of $D$ onto the unit $\ell_2$-ball after every update. The resulting loss curves for both GD and BDC are shown in Figure 4. As illustrated in the Figure 4, BDCA converges faster and attains a lower objective value compared to GD.

**Berkeley Segmentation Dataset. Martin et al. (2001)** From the BSDS500 training set (200 images), we randomly extract 50 grayscale patches of size $8 \times 8$ from each image. Any patch that is

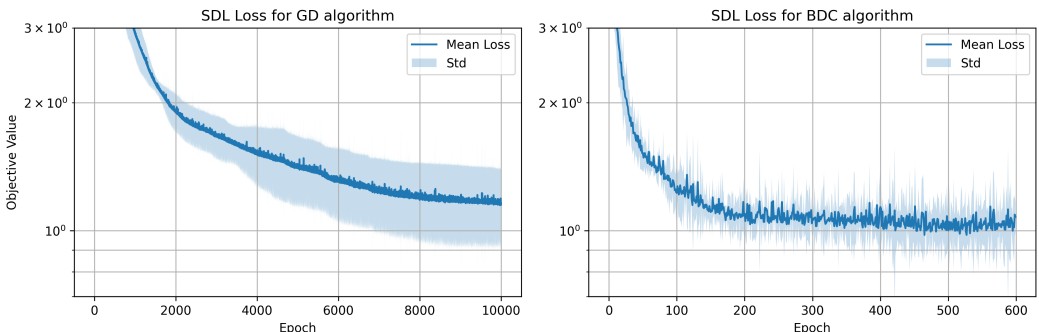

Figure 4: SDL loss evolution with the nonconvex $\ell_1 - \ell_Q$ regularizer (Eq. 5.5) on synthetic data. The left panel shows the loss trajectory of the joint full-batch gradient descent method with an adaptive step size, while the right panel shows the loss evolution of the BDC algorithm. Each curve reports the mean and $\pm 2$ standard deviations over 3 independent runs.

identically zero is discarded; the remaining patches are demeaned and normalized to unit $\ell_2$-norm, then assembled as columns of $Y$. For this experiment we use $\alpha = 0.2$, $Q = 5$, and $k = 256$. Results are shown in Figure 2.

### A.6.3 TRAINING NEURAL NETWORKS.

Here, we explain the implementation structures of the training problem with more detail.

**Implementation details (Regression Task).** For the regression task's training we set 50 epochs and a batch size of 20. The training network included three linear layers with sequential input-output dimensions $(13, 64, 32, 16, 1)$ and with ReLU activation functions. The training result was compared with SGD as a benchmark method with step-size $10^{-2}$. The BDC sub-problems were solved with 100 calls to the minimization oracle. Here, we used a constant $\rho = 10^3$. For the BDC subproblems, simple GD was utilized. The results for 10 Monte-Carlo instances and 68% confidence intervals are shown in Figure 3 (left).

**Implementation details (Classification Task).** For the classification task, we tested CIFAR10 dataset, and FASHIONMNIST datasets. For the FASHIONMNIST dataset, we considered a three layer ReLU network with sequential input-output dimensions $(28 * 28, 512, 64, 10)$. The training step-size for SGD was set to $10^{-2}$, the batch size was fixed to 256, and epoch is 100. The inner iterations for solving BDC sub-problems using GD was fixed to 100. The results for 10 Monte-Carlo instances, 90% confidence intervals, and $\rho = 1/3 \times 10^3$ are depicted in Figure 3 (middle).

For the CIFAR10 dataset, we considered a four layer ReLU network with sequential input-output dimensions $(3 * 32 * 32, 256, 128, 64, 10)$. The training step-size for the SGD method was set to $10^{-2}$, the batch size was fixed to 128, and the epoch is 100. The inner iterations for solving BDC sub-problems was fixed to 100 with a similar step-size strategy as for FASHIONMNIST. The results for 10 Monte-Carlo instances, 90% confidence intervals, and $\rho = 10^3$ are depicted in Figure 3 (right).

# B PROOFS

## B.1 PROOF OF PROPOSITION 2.1

Fix any block $i \in [n]$ and fix an arbitrary complement $\bar{\boldsymbol{\theta}}_i \in \bar{\mathcal{X}}_i$. We work with the $i$-th block (with $\bar{\boldsymbol{\theta}}_i$ fixed and $\theta_i$ free). By BDC assumption, for each $r \in \{1, \ldots, m\}$ there exist functions

$$g_i^{(r)}(\cdot\,;\bar{\boldsymbol{\theta}}_i),\ h_i^{(r)}(\cdot\,;\bar{\boldsymbol{\theta}}_i):\ \mathcal{X}_i \to \mathbb{R}$$

that are convex in $\theta_i$ such that

$$f_r(\boldsymbol{\theta}) = g_i^{(r)}(\theta_i;\bar{\boldsymbol{\theta}}_i) - h_i^{(r)}(\theta_i;\bar{\boldsymbol{\theta}}_i), \qquad \boldsymbol{\theta}_i \in \mathcal{X}_i.$$

We show that each operation preserves this BDC form.

**(i) Linear combinations.** Let $\alpha_1, \ldots, \alpha_m \in \mathbb{R}$ and write $\alpha_r = \alpha_r^+ - \alpha_r^-$ with $\alpha_r^\pm \geq 0$. Then, for every $\boldsymbol{\theta}_i \in \mathcal{X}_i$,

$$\sum_{r=1}^m \alpha_r\, f_r(\boldsymbol{\theta}) = \sum_{r=1}^m \alpha_r^+ \big(g_i^{(r)}(\theta_i;\bar{\boldsymbol{\theta}}_i) - h_i^{(r)}(\theta_i;\bar{\boldsymbol{\theta}}_i)\big)$$

$$- \sum_{r=1}^m \alpha_r^- \big(g_i^{(r)}(\theta_i;\bar{\boldsymbol{\theta}}_i) - h_i^{(r)}(\theta_i;\bar{\boldsymbol{\theta}}_i)\big)$$

$$= \underbrace{\Big( \sum_{r=1}^m \alpha_r^+\, g_i^{(r)}(\theta_i;\bar{\boldsymbol{\theta}}_i) + \sum_{r=1}^m \alpha_r^-\, h_i^{(r)}(\theta_i;\bar{\boldsymbol{\theta}}_i) \Big)}_{\text{convex in } \boldsymbol{\theta}_i}$$

$$- \underbrace{\Big( \sum_{r=1}^m \alpha_r^+\, h_i^{(r)}(\theta_i;\bar{\boldsymbol{\theta}}_i) + \sum_{r=1}^m \alpha_r^-\, g_i^{(r)}(\theta_i;\bar{\boldsymbol{\theta}}_i) \Big)}_{\text{convex in } \boldsymbol{\theta}_i}.$$

Each bracket is a nonnegative sum of convex functions of $\theta_i$, hence convex. Therefore $\sum_{r=1}^m \alpha_r f_r$ is BDC.

**(ii) Maximum.** Using the BDC decompositions of all $f_r$, for every $\theta_i \in \mathcal{X}_i$,

$$\max_{1 \leq r \leq m} f_r(\boldsymbol{\theta}) = \max_{1 \leq r \leq m} \Big\{ g_i^{(r)}(\theta_i;\bar{\boldsymbol{\theta}}_i) - h_i^{(r)}(\theta_i;\bar{\boldsymbol{\theta}}_i) \Big\}$$

$$= \max_{1 \leq r \leq m} \Big\{ g_i^{(r)}(\theta_i;\bar{\boldsymbol{\theta}}_i) + \sum_{\substack{s=1 \\ s \neq r}}^m h_i^{(s)}(\theta_i;\bar{\boldsymbol{\theta}}_i) \Big\} - \sum_{k=1}^m h_i^{(k)}(\theta_i;\bar{\boldsymbol{\theta}}_i). \qquad \text{(B.1)}$$

For the fixed $\bar{\boldsymbol{\theta}}_i$, each inner map

$$\theta_i \ \mapsto\ g_i^{(r)}(\theta_i;\bar{\boldsymbol{\theta}}_i) + \sum_{\substack{s=1 \\ s \neq r}}^m h_i^{(s)}(\theta_i;\bar{\boldsymbol{\theta}}_i)$$

is convex in $\theta_i$ (sum of convex functions); the pointwise maximum over finitely many convex functions is convex in $\theta_i$; and the final sum $\sum_{k=1}^m h_i^{(k)}(\theta_i;\bar{\boldsymbol{\theta}}_i)$ is convex in $\theta_i$. Hence the right-hand side of (B.1) is a difference of two convex functions of $\theta_i$, proving that $\max_r f_r$ is BDC.

**(iii) Minimum.** By part (i) with $\alpha_r = -1$, the function $-f_r$ is BDC for each $r$. Applying part (ii) to $\{-f_r\}_{r=1}^m$ and using

$$\min_{1 \leq r \leq m} f_r(\boldsymbol{\theta}) = - \max_{1 \leq r \leq m} \big( -f_r(\boldsymbol{\theta}) \big),$$

we conclude that $\min_r f_r$ is BDC.

Since the block $i$ was arbitrary, all three operations preserve the BDC property.

## B.2 PROOF OF THEOREM 3.1

We prove the proposition by treating the *even-degree* and *odd-degree* cases separately. In each case we first derive an *upper bound* via the polarization identity of monomials together with a precise pairing argument that halves the raw atom count, and then obtain a *lower bound* by relating any DC decomposition to a (real) Waring decomposition and invoking known rank formulas for monomials.

**Preliminaries**

*Polarization identity of monomials.* We use the following polynomial identity.

**Lemma B.1** (Polarization identity of monomials Kan (2008)). *Let* $b_1, \ldots, b_M \in \mathbb{Z}_{\geq 0}$ *with* $S = \sum_{i=1}^{M} b_i$ *and variables* $\theta_1, \ldots, \theta_M$. *Then*

$$\prod_{i=1}^{M} \theta_i^{b_i} = \frac{1}{S!} \sum_{v_1=0}^{b_1} \cdots \sum_{v_M=0}^{b_M} (-1)^{\sum_{i=1}^{M} v_i} \prod_{i=1}^{M} \binom{b_i}{v_i} \left( \sum_{i=1}^{M} (\tfrac{b_i}{2} - v_i)\theta_i \right)^{S}.$$

*Waring decompositions and ranks.* A *Waring decomposition* of a degree-$S$ homogeneous polynomial (form) $F$ is an identity

$$F(\boldsymbol{\theta}) = \sum_{j=1}^{r} c_j \, \ell_j(\boldsymbol{\theta})^{S}, \qquad \ell_j(\boldsymbol{\theta}) = a_{j1}\theta_1 + \cdots + a_{jm}\theta_n.$$

We distinguish two notions:

- **Complex Waring rank** $\mathrm{rk}_{\mathbb{C}}(F)$: the minimal $r$ for which there exist real scalars $c_j$ and linear forms $\ell_j$ with complex coefficients such that $F = \sum_{j=1}^{r} c_j \, \ell_j^{S}$.

- **Real Waring rank** $\mathrm{rk}_{\mathbb{R}}(F)$: the minimal $r$ for which there exist *real* $c_j$ and *real-coefficient* $\ell_j$ such that $F = \sum_{j=1}^{r} c_j \, \ell_j^{S}$.

Allowing complex coefficients cannot increase the minimum, hence

$$\mathrm{rk}_{\mathbb{C}}(F) \; \leq \; \mathrm{rk}_{\mathbb{R}}(F).$$

**Case 1:** $s$ **even (linear-even atoms** $(u^{\top}\boldsymbol{\theta})^s$**).**

First, we upper bound $N$ via the polarization identity. Apply Lemma B.1 with $M = n$, $S = s$, $z_i = \theta_i$. This expresses $f$ as a linear combination of $\prod_{i=1}^{n}(b_i + 1)$ degree-$s$ powers of linear forms:

$$f(\boldsymbol{\theta}) = \frac{1}{s!} \sum_{v_1=0}^{b_1} \cdots \sum_{v_n=0}^{b_n} (-1)^{\sum_{i=1}^{n} v_i} \left( \prod_{i=1}^{n} \binom{b_i}{v_i} \right) \left( \sum_{i=1}^{n} \left(\tfrac{b_i}{2} - v_i\right)\theta_i \right)^{s}.$$

Write $v = (v_1, \ldots, v_n)$ and let the *complement* be $b - v = (b_1 - v_1, \ldots, b_n - v_n)$. Since $s$ is even, we have

$$\left( \sum_{i=1}^{n} \left(\tfrac{b_i}{2} - (b_i - v_i)\right)\theta_i \right)^{s} = \left( -\sum_{i=1}^{n} \left(\tfrac{b_i}{2} - v_i\right)\theta_i \right)^{s} = \left( \sum_{i=1}^{n} \left(\tfrac{b_i}{2} - v_i\right)\theta_i \right)^{s},$$

so the two atoms coincide. Thus each complementary pair $\{v, b - v\}$ contributes *twice* the same atom, and pairing halves the count to $\frac{1}{2} \prod_{i=1}^{n}(b_i + 1)$. If all $b_i$ are even, when $v_i = b_i/2$, we have $\frac{b_i}{2} - v_i = 0$, so the exact number of nonzero atoms equals $\left( \prod_{i=1}^{n}(b_i+1) - 1 \right)/2 = \left\lfloor \frac{1}{2} \prod_{i=1}^{n}(b_i+1) \right\rfloor$. Therefore,

$$N \; \leq \; \left\lfloor \tfrac{1}{2} \prod_{i=1}^{n}(b_i + 1) \right\rfloor.$$

Second, we find the lower bound $N$ via Waring rank. Any DC split in this model can be written as

$$f(\boldsymbol{\theta}) = g(\boldsymbol{\theta}) - h(\boldsymbol{\theta}) = \sum_{i=1}^{r} \alpha_i \, (u_i^{\top}\boldsymbol{\theta})^s \; - \sum_{i=r+1}^{r+q} \alpha_i \, (u_i^{\top}\boldsymbol{\theta})^s = \sum_{i=1}^{N} c_i \, (u_i^{\top}\boldsymbol{\theta})^s,$$

with $\alpha_i > 0$, $c_i = \pm\alpha_i$, and $N = r + q$. This is a Waring decomposition with real coefficients and real linear forms, hence

$$N \geq \mathrm{rk}_{\mathbb{R}}(f) \geq \mathrm{rk}_{\mathbb{C}}(f).$$

For monomials the complex rank is known exactly:

**Lemma B.2** (Complex Waring rank of a monomial Carlini et al. (2012)). *Let $b_1, \ldots, b_n \in \mathbb{Z}_{\geq 0}$ with $s = \sum_{i=1}^{n} b_i$, and $1 \leq b_1 \leq \cdots \leq b_n$. For the monomial $f(\boldsymbol{\theta}) = \theta_1^{b_1} \cdots \theta_n^{b_n}$,*

$$\mathrm{rk}_{\mathbb{C}}(f) = \prod_{i=2}^{n} (b_i + 1).$$

Combining $N \geq \mathrm{rk}_{\mathbb{C}}(f)$ with Lemma B.2 yields

$$N \geq \prod_{i=2}^{n} (b_i + 1).$$

Finally, the relationship between real and complex ranks clarifies tightness:

**Theorem B.3** (Carlini et al. (2017)). *Let $f(\boldsymbol{\theta}) = \theta_1^{b_1} \cdots \theta_n^{b_n}$ with $1 \leq b_1 \leq \cdots \leq b_n$. Then $\mathrm{rk}_{\mathbb{R}}(f) = \mathrm{rk}_{\mathbb{C}}(f)$ if and only if $b_1 = 1$.*

Hence, when $b_1 = 1$ the lower bound $\prod_{i=2}^{n}(b_i + 1)$ equals the complex (and real) rank, and together with the polarization upper bound we obtain matching bounds; for $b_1 > 1$ a strict gap can remain.

**Case 2: $s$ odd (affine-even atoms $(u^\top \boldsymbol{\theta} + d)^{s+1}$).** We now handle $s$ odd, where $(u^\top \boldsymbol{\theta})^s$ is not convex. To remain in a convex-atom setting we use *even-degree affine atoms*, obtained via degree-$d = s + 1$ homogenization.

Define

$$F(\boldsymbol{\theta}, t) = t\, f(\boldsymbol{\theta}) = t\, \theta_1^{b_1} \cdots \theta_n^{b_n},$$

which is homogeneous of even degree $S = s + 1$ in the variables $(\boldsymbol{\theta}, t) \in \mathbb{R}^n \times \mathbb{R}$. Any atom of the form $(u^\top \boldsymbol{\theta} + d\,t)^S$ is convex (even power of an affine form). Evaluating any homogeneous decomposition of $F$ at $t = 1$ yields atoms $(u^\top \boldsymbol{\theta} + d)^{s+1}$, which remain convex in $\boldsymbol{\theta}$.

Now, we upper bound $N$ via the polarization identity and pairing. Apply Lemma B.1 to the $(n+1)$-variate monomial $t\, \theta_1^{b_1} \cdots \theta_n^{b_n}$ with $(z_0, \ldots, z_n) = (t, \theta_1, \ldots, \theta_n)$, $(b_0, \ldots, b_n) = (1, b_1, \ldots, b_n)$, and $S = s + 1$. We obtain

$$F(\boldsymbol{\theta}, t) = \frac{1}{S!} \sum_{v_0=0}^{1} \sum_{v_1=0}^{b_1} \cdots \sum_{v_n=0}^{b_n} (-1)^{\sum_{i=0}^{n} v_i} \left( \prod_{i=0}^{n} \binom{b_i}{v_i} \right) \left( \sum_{i=0}^{n} \left( \frac{b_i}{2} - v_i \right) z_i \right)^{S},$$

a signed sum of even powers of affine forms $(d\,t + u^\top \boldsymbol{\theta})^S$. Pair each index $v = (v_0, \ldots, v_n)$ with its complement $b - v$. Hence the raw count $(b_0 + 1) \prod_{i=1}^{n} (b_i + 1) = 2 \prod_{i=1}^{n} (b_i + 1)$ collapses *exactly* by a factor 2, yielding

$$\#\text{atoms in } F = \prod_{i=1}^{n} (b_i + 1).$$

Setting $t = 1$ gives a DC decomposition of $f$ with convex affine-power atoms $(u^\top \boldsymbol{\theta} + d)^{s+1}$ and

$$N \leq \prod_{i=1}^{n} (b_i + 1).$$

*Affine vs. homogeneous decompositions.* In the odd-degree case we use degree-$d = s + 1$ powers of *affine* forms $(\ell(\boldsymbol{\theta}) + \beta)^d$. It is crucial that sums of such affine powers correspond *exactly* to homogeneous sums of degree-$d$ powers of linear forms in one extra variable, with a *term-by-term* correspondence that preserves the number of terms.

**Lemma B.4** (Affine–homogeneous correspondence). *Let $f : \mathbb{R}^n \to \mathbb{R}$ be of degree $d$, and let its degree-$d$ homogenization be $F(X_0, X) = X_0^d f(X/X_0)$, so that $F$ is homogeneous of degree $d$ and $F(1, \boldsymbol{\theta}) = f(\boldsymbol{\theta})$. Then the following are equivalent:*

1. $f(\boldsymbol{\theta}) = \sum_{j=1}^{r} c_j \left( \ell_j(\boldsymbol{\theta}) + \beta_j \right)^d$ *(affine sum of degree-d powers)*.

2. $F(X_0, X) = \sum_{j=1}^{r} c_j \left( \beta_j X_0 + \ell_j(X) \right)^d$ *(homogeneous sum of degree-d powers)*.

*Moreover, the number of terms $r$ is preserved in both directions.*

*Proof.* (1) $\Rightarrow$ (2): Substitute $\boldsymbol{\theta} = X/X_0$ and multiply by $X_0^d$, then expand: $F(X_0, X) = X_0^d f(X/X_0) = \sum_j c_j \left( \beta_j X_0 + \ell_j(X) \right)^d$. (2) $\Rightarrow$ (1): Evaluate at $X_0 = 1$ to get $f(\boldsymbol{\theta}) = F(1, \boldsymbol{\theta}) = \sum_j c_j \left( \ell_j(\boldsymbol{\theta}) + \beta_j \right)^d$. Thus the atoms correspond bijectively and the count $r$ is unchanged. $\square$

By Lemma B.4 with $d = S = s + 1$, every DC decomposition of $f$ into affine atoms $(\ell(\boldsymbol{\theta}) + \beta)^{s+1}$ induces a homogeneous Waring decomposition of $F$ into atoms $(\beta t + \ell(\boldsymbol{\theta}))^{s+1}$ with the *same* number of terms, and conversely any homogeneous decomposition of $F$ restricts at $t = 1$ to a decomposition of $f$ with the *same* number of terms. Thus the minimal atom count in our odd-$s$ DC model equals the affine Waring rank of $f$ at degree $s + 1$, which by Lemma B.4 equals the Waring rank of $F$.

So we lower bound $N$ via Waring rank of the lifted monomial. Any such DC decomposition of $f$ induces a decomposition of $F$ of the form

$$F(\boldsymbol{\theta}, t) = \sum_{j=1}^{N} c_j \left( u_j^\top \boldsymbol{\theta} + d_j t \right)^{s+1},$$

which is a real Waring decomposition of the $(n+1)$-variate monomial $t^1 \theta_1^{b_1} \cdots \theta_n^{b_n}$ of degree $S = s + 1$. The complex Waring rank of this monomial equals

$$\mathrm{rk}_{\mathbb{C}}(t^1 \theta_1^{b_1} \cdots \theta_n^{b_n}) = \prod_{i=1}^{n} (b_i + 1),$$

and, since the smallest exponent is 1, real and complex ranks coincide (see Theorem B.3). Therefore

$$N \geq \mathrm{rk}_{\mathbb{R}}(F) = \mathrm{rk}_{\mathbb{C}}(F) = \prod_{i=1}^{n} (b_i + 1).$$

Together with the upper bound we conclude

$$N = \prod_{i=1}^{n} (b_i + 1).$$

**Conclusion.** For even $s$, the polarization identity and complementary-index pairing yield $N \leq \left\lfloor \frac{1}{2} \prod_{i=1}^{n} (b_i + 1) \right\rfloor$, while the Waring-rank argument gives $N \geq \prod_{i=2}^{n} (b_i + 1)$; when $b_1 = 1$ these bounds are tight. For odd $s$, the degree-$(s+1)$ homogenization $F(\boldsymbol{\theta}, t) = t f(\boldsymbol{\theta})$, the polarization identity in $n+1$ variables, and the corresponding Waring-rank lower bound match exactly, giving $N = \prod_{i=1}^{n} (b_i + 1)$.

### B.3 PROOF OF THEOREM 3.2

Fix an arbitrary block $\theta_l$ and hold all other blocks fixed. When we refer to 'convex' for a vector-valued function in the proof, we mean it in the componentwise sense. We consider two cases.

**Case 1: Hidden block $\theta_l = (W_l, b_l)$.**

First, We prove by induction on $k$ that $Z_k^{\pm}$ are componentwise convex in $(W_l, b_l)$ and satisfy $Z_k^{\pm} \geq 0$.

*Base ($k < l$):* For $k < l$, the quantities $Z_k^{\pm}$ do not depend on $(W_l, b_l)$ and are thus constant (hence convex) w.r.t. $(W_l, b_l)$. It remains to justify nonnegativity for these layers. at $k = 1$,

$$Z_1^+ = \sigma(W_1 x + b_1) \geq 0, \qquad Z_1^- = 0.$$

Assume $Z_s^{\pm} \geq 0$ for some $s < l - 1$. Then

$$Z_{s+1}^- = \sigma(W_{s+1}) Z_s^- + \sigma(-W_{s+1}) Z_s^+ \geq 0,$$

because $\sigma(W_{s+1})$ and $\sigma(-W_{s+1})$ are entrywise nonnegative. Moreover,

$$Z_{s+1}^+ = \max\Big\{ \underbrace{\sigma(W_{s+1})Z_s^+ + \sigma(-W_{s+1})Z_s^- + b_{s+1}}_{p_{s+1}}, \; Z_{s+1}^- \Big\} \; \geq \; Z_{s+1}^- \; \geq \; 0.$$

By induction, $Z_k^\pm \geq 0$ for all $k < l$.

*Layer $k = l$:* With $Z_{l-1}^\pm \geq 0$ fixed,

$$p_l = \sigma(W_l)Z_{l-1}^+ + \sigma(-W_l)Z_{l-1}^- + b_l, \quad Z_l^- = \sigma(W_l)Z_{l-1}^- + \sigma(-W_l)Z_{l-1}^+.$$

Entrywise $w \mapsto \sigma(\pm w)$ are convex and nonnegative; multiplying by fixed nonnegative vectors $Z_{l-1}^\pm$ and adding the affine term $b_l$ preserve convexity. Hence $p_l$ and $Z_l^-$ are convex, with $Z_l^- \geq 0$. Set

$$Z_l^+ \; = \; \max\{p_l, Z_l^-\},$$

which is convex (pointwise max preserves convexity) and satisfies $Z_l^+ \geq Z_l^- \geq 0$.

*Induction ($k \to k+1$ for $k \geq l$):* Assume $Z_k^\pm$ are convex in $(W_l, b_l)$ and $Z_k^\pm \geq 0$. For fixed $(W_{k+1}, b_{k+1})$, the matrices $\sigma(W_{k+1})$ and $\sigma(-W_{k+1})$ are entrywise nonnegative constants. Thus

$$p_{k+1} = \sigma(W_{k+1})Z_k^+ + \sigma(-W_{k+1})Z_k^- + b_{k+1}, \quad Z_{k+1}^- = \sigma(W_{k+1})Z_k^- + \sigma(-W_{k+1})Z_k^+$$

are nonnegative linear images of $(Z_k^+, Z_k^-)$ plus a constant; hence $Z_{k+1}^- \geq 0$ and both $p_{k+1}, Z_{k+1}^-$ are convex. Finally,

$$Z_{k+1}^+ = \max\{p_{k+1}, Z_{k+1}^-\}$$

is convex and satisfies $Z_{k+1}^+ \geq Z_{k+1}^- \geq 0$. By induction, this holds for all $k \geq l$, in particular for $k = L - 1$. Using $\sigma(a - b) = \max\{a, b\} - b$ coordinatewise,

$$Z_{k+1}^+ - Z_{k+1}^- \; = \; \sigma\big(W_{k+1}(Z_k^+ - Z_k^-) + b_{k+1}\big),$$

so $a_{k+1} = Z_{k+1}^+ - Z_{k+1}^-$ and in particular $a_{L-1} = Z_{L-1}^+ - Z_{L-1}^-$.

at the end, keep $(W_L, b_L)$ fixed. For each class $c$,

$$A_c(\boldsymbol{\theta}) = \sigma(W_{L,c})Z_{L-1}^+ + \sigma(-W_{L,c})Z_{L-1}^- + \sigma(b_{L,c}), \quad B_c(\boldsymbol{\theta}) = \sigma(W_{L,c})Z_{L-1}^- + \sigma(-W_{L,c})Z_{L-1}^+ + \sigma(-b_{L,c}).$$

Here $\sigma(\pm W_{L,c}) \geq 0$ and $\sigma(\pm b_{L,c}) \geq 0$ are constants; therefore $A_c(\cdot\,; \bar{\boldsymbol{\theta}}_l), B_c(\cdot\,; \bar{\boldsymbol{\theta}}_l)$ are nonnegative linear combinations of the convex functions $Z_{L-1}^\pm$ plus constants, hence are convex and nonnegative in $\theta_l = (W_l, b_l)$. Using $\sigma(t) - \sigma(-t) = t$ entrywise and $a_{L-1} = Z_{L-1}^+ - Z_{L-1}^-$, we obtain

$$A(\boldsymbol{\theta}) - B(\boldsymbol{\theta}) = \big(\sigma(W_L) - \sigma(-W_L)\big)(Z_{L-1}^+ - Z_{L-1}^-) + \big(\sigma(b_L) - \sigma(-b_L)\big) = W_L a_{L-1} + b_L = F_x(\boldsymbol{\theta}).$$

**Case 2: Output block $\theta_L = (W_L, b_L)$.**

Here $Z_{L-1}^\pm$ are fixed and nonnegative. The entrywise maps $W_L \mapsto \sigma(\pm W_L)$ and $b_L \mapsto \sigma(\pm b_L)$ are convex and nonnegative. Hence each component of $A(\cdot\,; \bar{\boldsymbol{\theta}}_L), B(\cdot\,; \bar{\boldsymbol{\theta}}_L)$ in (3.1) is a nonnegative linear combination of nonnegative convex functions, and is therefore convex and nonnegative in $(W_L, b_L)$.

B.4    PROOF OF PROPOSITION 3.4

Fix an arbitrary block $i \in [n]$ and fix $\bar{\boldsymbol{\theta}}_i$. By the componentwise BDC assumption, for each coordinate $j = 1, \ldots, m$ there exist convex functions $a_{ij}(\cdot\,; \bar{\boldsymbol{\theta}}_i)$ and $b_{ij}(\cdot\,; \bar{\boldsymbol{\theta}}_i)$ in $\theta_i$ such that $E_j(\boldsymbol{\theta}) = a_{ij}(\theta_i; \bar{\boldsymbol{\theta}}_i) - b_{ij}(\theta_i; \bar{\boldsymbol{\theta}}_i)$.

*Multi-Block convexity of $g$.* From the conjugate definition and $E_j = a_{ij} - b_{ij}$,

$$f^*(E(\boldsymbol{\theta})) = \max_{u \in U}\Big\{ \langle u, a_i(\theta_i; \bar{\boldsymbol{\theta}}_i)\rangle - \langle u, b_i(\theta_i; \bar{\boldsymbol{\theta}}_i)\rangle - f(u)\Big\}.$$

Adding $h_i$ yields the variational form

$$g_i(\theta_i; \bar{\boldsymbol{\theta}}_i) = \max_{u \in U}\Big\{ \langle u + c^+, a_i(\theta_i; \bar{\boldsymbol{\theta}}_i)\rangle + \langle -u + d^+, b_i(\theta_i; \bar{\boldsymbol{\theta}}_i)\rangle - f(u)\Big\}.$$

For any fixed $u \in U$, the map

$$\theta_i \mapsto \langle u + c^+, a_i(\theta_i; \bar{\boldsymbol{\theta}}_i) \rangle + \langle -u + d^+, b_i(\theta_i; \bar{\boldsymbol{\theta}}_i) \rangle - f(u)$$

is convex in $\theta_i$ since $u_j + c_j^+ \geq 0$ and $-u_j + d_j^+ \geq 0$ for all $j$, making it a nonnegative linear combination of convex functions. Taking the pointwise maximum over $u \in U$ preserves convexity, so $g_i(\cdot; \bar{\boldsymbol{\theta}}_i)$ is convex.

*Multi-Block convexity of $h$.* By definition,

$$h_i(\theta_i; \bar{\boldsymbol{\theta}}_i) = \langle c^+, a_i(\theta_i; \bar{\boldsymbol{\theta}}_i) \rangle + \langle d^+, b_i(\theta_i; \bar{\boldsymbol{\theta}}_i) \rangle,$$

which is a nonnegative linear combination of convex functions of $\theta_i$, hence convex.

Finally, by construction,

$$f^*(E(\boldsymbol{\theta})) = g_i(\theta_i; \bar{\boldsymbol{\theta}}_i) - h_i(\theta_i; \bar{\boldsymbol{\theta}}_i),$$

so $f^* \circ E$ admits a multi-block DC decomposition. Since block $i$ was arbitrary, $f^* \circ E$ is BDC.

### B.5 PROOF OF LEMMA A.3

Due to (4.4), we have

$$u_{i_k}^k \in \partial h_{i_k}(\theta_{i_k}^k; \bar{\boldsymbol{\theta}}_{i_k}^k) \quad \text{and} \quad \langle u_{i_k}^k, \theta_{i_k}^k - \theta_{i_k}^{k+1} \rangle \leq g_{i_k}(\theta_{i_k}^k; \bar{\boldsymbol{\theta}}_{i_k}^k) - g_{i_k}(\theta_{i_k}^{k+1}; \bar{\boldsymbol{\theta}}_{i_k}^k) - \frac{\rho}{2} \|\theta_{i_k}^{k+1} - \theta_{i_k}^k\|^2. \tag{B.2}$$

Using convexity of $g_{i_k}(\cdot, \bar{\boldsymbol{\theta}}_{i_k}^k)$ in (B.2), we have

$$\langle u_{i_k}^k, \theta_{i_k}^k - \theta_{i_k}^{k+1} \rangle \leq g_{i_k}(\theta_{i_k}^k; \bar{\boldsymbol{\theta}}_{i_k}^k) - g_{i_k}(\theta_{i_k}^{k+1}; \bar{\boldsymbol{\theta}}_{i_k}^k) - \frac{\rho}{2} \|\theta_{i_k}^{k+1} - \theta_{i_k}^k\|^2,$$

$$\implies \langle u_{i_k}^k, \theta_{i_k}^k - \theta_{i_k}^{k+1} \rangle \leq -\langle \nabla g_{i_k}(\theta_{i_k}^k; \bar{\boldsymbol{\theta}}_{i_k}^k), \theta_{i_k}^{k+1} - \theta_{i_k}^k \rangle - \frac{\rho}{2} \|\theta_{i_k}^{k+1} - \theta_{i_k}^k\|^2,$$

$$\implies \frac{\rho}{2} \|\theta_{i_k}^{k+1} - \theta_{i_k}^k\|^2 \leq \langle \nabla g_{i_k}(\theta_{i_k}^k; \bar{\boldsymbol{\theta}}_{i_k}^k) - u_{i_k}^k, \theta_{i_k}^k - \theta_{i_k}^{k+1} \rangle,$$

$$\implies \frac{\rho}{2} \|\theta_{i_k}^{k+1} - \theta_{i_k}^k\|^2 \leq \mathcal{G}(\boldsymbol{\theta}^k) \|\theta_{i_k}^{k+1} - \theta_{i_k}^k\|,$$

$$\implies \|\boldsymbol{\theta}^{k+1} - \boldsymbol{\theta}^k\| \leq \frac{2}{\rho} \mathcal{G}(\boldsymbol{\theta}^k).$$

We get that the update $\boldsymbol{\theta}^{k+1}$ is in $\mathcal{B}\left(\boldsymbol{\theta}^k, \frac{2}{\rho}\mathcal{G}(\boldsymbol{\theta}^k)\right)$.

### B.6 PROOF OF LEMMA A.4

From (4.4) we know:

$$u_{i_k}^k \in \partial h_{i_k}(\theta_{i_k}^k; \bar{\boldsymbol{\theta}}_{i_k}^k) \quad \text{and} \quad \langle u_{i_k}^k, \theta_{i_k}^k - \theta_{i_k}^{k+1} \rangle \leq g_{i_k}(\theta_{i_k}^k; \bar{\boldsymbol{\theta}}_{i_k}^k) - g_{i_k}(\theta_{i_k}^{k+1}; \bar{\boldsymbol{\theta}}_{i_k}^k) - \frac{\rho}{2} \|\theta_{i_k}^{k+1} - \theta_{i_k}^k\|^2.$$

Now, using convexity of $h_{i_k}$ we get:

$$h_{i_k}(\theta_{i_k}^k; \bar{\boldsymbol{\theta}}_{i_k}^k) - h_{i_k}(\theta_{i_k}^{k+1}; \bar{\boldsymbol{\theta}}_{i_k}^k) \leq g_{i_k}(\theta_{i_k}^k; \bar{\boldsymbol{\theta}}_{i_k}^k) - g_{i_k}(\theta_{i_k}^{k+1}; \bar{\boldsymbol{\theta}}_{i_k}^k) - \frac{\rho}{2} \|\theta_{i_k}^{k+1} - \theta_{i_k}^k\|^2,$$

$$g_{i_k}(\theta_{i_k}^{k+1}; \bar{\boldsymbol{\theta}}_{i_k}^k) \leq g_{i_k}(\theta_{i_k}^k; \bar{\boldsymbol{\theta}}_{i_k}^k) + h_{i_k}(\theta_{i_k}^{k+1}; \bar{\boldsymbol{\theta}}_{i_k}^k) - h_{i_k}(\theta_{i_k}^k; \bar{\boldsymbol{\theta}}_{i_k}^k),$$

$$f(\boldsymbol{\theta}^{k+1}) \leq f(\boldsymbol{\theta}^k)$$

Unrolling this inequality to the initialization gives

$$f(\boldsymbol{\theta}^{k+1}) \leq f(\boldsymbol{\theta}^0)$$

$$g_{i_k}(\theta_{i_k}^{k+1}; \bar{\boldsymbol{\theta}}_{i_k}^k) \leq g_{i_0}(\theta_{i_0}^0; \bar{\boldsymbol{\theta}}_{i_0}^0) + h_{i_k}(\theta_{i_k}^{k+1}; \bar{\boldsymbol{\theta}}_{i_k}^k) - h_{i_0}(\theta_{i_0}^0; \bar{\boldsymbol{\theta}}_{i_0}^0),$$

$$\leq g_{i_0}(\theta_{i_0}^0; \bar{\boldsymbol{\theta}}_{i_0}^0) + H,$$

where we have used $h_{i_k}(\theta_{i_k}^{k+1}; \bar{\boldsymbol{\theta}}_{i_k}^k) - h_{i_0}(\theta_{i_0}^0; \bar{\boldsymbol{\theta}}_{i_0}^0) \leq H$ and the fact that $\bar{\theta}_{i_k}^{k+1} = \bar{\theta}_{i_k}^k$. Since this result holds for any $k$, through Corollary A.11 we have $\|\nabla g_{i_k}(\theta_{i_k}^k; \bar{\boldsymbol{\theta}}_{i_k}^k)\| \leq E$. Recall that $g_{i_k}$ is

$\ell$-smooth with a subquadratic $\ell$. Using Proposition A.10, $g_{i_k}$ is also $(r, m)$-smooth with $r(u) = \frac{a}{m(u)}$ and $m(u) := \ell(u + a)$ for some $a > 0$. Therefore, we can use Lemma A.9 if we ensure the updates are inside $\mathcal{B}(\boldsymbol{\theta}^k, r(E))$. Similar to the proof of Lemma A.3 (Appendix B.5), we know

$$\|\boldsymbol{\theta}^{k+1} - \boldsymbol{\theta}^k\| \leq \sup_{u \in \partial h_{i_k}(\theta_{i_k}^k; \bar{\boldsymbol{\theta}}_{i_k}^k)} \frac{2(\|\nabla g_{i_k}(\theta_{i_k}^k; \bar{\boldsymbol{\theta}}_{i_k}^k)\| + \|u\|)}{\rho} \leq \frac{2(E + R)}{\rho}.$$

As a result taking $\rho \geq \frac{2(E+R)}{r(E)} = \ell(2E)\frac{2(E+R)}{E}$ will satisfy the conditions in Lemma A.9. This implies the desired result.

## B.7 PROOF OF PROPOSITION A.5

Using the assumptions in the theorem statement, we know that for any $\boldsymbol{\theta}^k \in \mathcal{X}$, the update $\boldsymbol{\theta}^{k+1} \in \mathcal{B}(\boldsymbol{\theta}^k, r(E))$. For any $\theta_{i_k} \in \mathcal{B}(\theta_{i_k}^k, r(E))$, consider the surrogate function

$$\hat{f}(\theta_{i_k}; \bar{\boldsymbol{\theta}}_{i_k}^k) := g_{i_k}(\theta_{i_k}; \bar{\boldsymbol{\theta}}_{i_k}^k) - h_{i_k}(\theta_{i_k}^k; \bar{\boldsymbol{\theta}}_{i_k}^k) - \langle u_{i_k}^k, \theta_{i_k} - \theta_{i_k}^k \rangle + \frac{\rho}{2}\|\theta_{i_k}^k - \theta_{i_k}\|^2.$$

where $u_{i_k}^k \in \partial h_{i_k}(\theta_{i_k}^k; \bar{\boldsymbol{\theta}}_{i_k}^k)$. From (4.4) we know that

$$\hat{f}(\theta_{i_k}^{k+1}; \bar{\boldsymbol{\theta}}_{i_k}^k) \leq \hat{f}(\theta_{i_k}^k; \bar{\boldsymbol{\theta}}_{i_k}^k),$$

and further considering the descent Lemma A.9 for $g_{i_k}$ with $L = \ell(2E)$, we get

$$\hat{f}(\theta_{i_k}^{k+1}; \bar{\boldsymbol{\theta}}_{i_k}^k) \leq g_{i_k}(\theta_{i_k}^k; \bar{\boldsymbol{\theta}}_{i_k}^k) + \langle \nabla g_{i_k}(\theta_{i_k}^k; \bar{\boldsymbol{\theta}}_{i_k}^k), \theta_{i_k} - \theta_{i_k}^k \rangle + \frac{L+\rho}{2}\|\theta_{i_k} - \theta_{i_k}^k\|^2 - h_{i_k}(\theta_{i_k}^k; \bar{\boldsymbol{\theta}}_{i_k}^k) - \langle u_{i_k}^k, \theta_{i_k} - \theta_{i_k}^k \rangle.$$

By Assumption 1, we get

$$\hat{f}(\theta_{i_k}^{k+1}; \bar{\boldsymbol{\theta}}_{i_k}^k) \leq f(\boldsymbol{\theta}^k) + \langle \nabla g_{i_k}(\theta_{i_k}^k; \bar{\boldsymbol{\theta}}_{i_k}^k) - u_{i_k}^k, \theta_{i_k} - \theta_{i_k}^k \rangle + \frac{L+\rho}{2}\|\theta_{i_k} - \theta_{i_k}^k\|^2,$$

$$\langle \nabla g_{i_k}(\theta_{i_k}^k; \bar{\boldsymbol{\theta}}_{i_k}^k) - u_{i_k}^k, \theta_{i_k}^k - \theta_{i_k} \rangle \leq f(\boldsymbol{\theta}^k) + \frac{L+\rho}{2}\|\theta_{i_k} - \theta_{i_k}^k\|^2 - \hat{f}(\theta_{i_k}^{k+1}; \bar{\boldsymbol{\theta}}_{i_k}^k),$$

$$\langle \nabla g_{i_k}(\theta_{i_k}^k; \bar{\boldsymbol{\theta}}_{i_k}^k) - u_{i_k}^k, \theta_{i_k}^k - \theta_{i_k} \rangle \leq g_{i_k}(\theta_{i_k}^k; \bar{\boldsymbol{\theta}}_{i_k}^k) - g_{i_k}(\theta_{i_k}^{k+1}; \bar{\boldsymbol{\theta}}_{i_k}^k) + \frac{L+\rho}{2}\|\theta_{i_k} - \theta_{i_k}^k\|^2 - \frac{\rho}{2}\|\theta_{i_k}^{k+1} - \theta_{i_k}^k\|^2$$
$$+ \langle u_{i_k}^k, \theta_{i_k}^{k+1} - \theta_{i_k}^k \rangle,$$

$$\text{(B.3)}$$

Note that if we choose $\rho = \frac{2(E+R)}{r(E)}$, then we know that

$$\frac{\rho}{2}\|\theta_{i_k} - \theta_{i_k}^k\|^2 - \frac{\rho}{2}\|\theta_{i_k}^{k+1} - \theta_{i_k}^k\|^2 \leq 0,$$

since in this case $\frac{\rho}{2}\|\theta_{i_k}^{k+1} - \theta_{i_k}^k\|^2 = r(E)$. However, in the more general case of $\rho \geq \frac{2(E+R)}{r(E)}$, this may not hold. Here, we proceed with the general case. Using the negetavity of $-\frac{\rho}{2}\|\theta_{i_k}^{k+1} - \theta_{i_k}^k\|^2 \leq 0$, we have

$$\langle \nabla g_{i_k}(\theta_{i_k}^k; \bar{\boldsymbol{\theta}}_{i_k}^k) - u_{i_k}^k, \theta_{i_k}^k - \theta_{i_k} \rangle - \frac{L+\rho}{2}\|\theta_{i_k} - \theta_{i_k}^k\|^2 \leq g_{i_k}(\theta_{i_k}^k; \bar{\boldsymbol{\theta}}_{i_k}^k) - g_{i_k}(\theta_{i_k}^{k+1}; \bar{\boldsymbol{\theta}}_{i_k}^k) + \langle u_{i_k}^k, \theta_{i_k}^{k+1} - \theta_{i_k}^k \rangle,$$

$$\langle \nabla g_{i_k}(\theta_{i_k}^k; \bar{\boldsymbol{\theta}}_{i_k}^k) - u_{i_k}^k, \theta_{i_k}^k - \theta_{i_k} \rangle - \frac{L+\rho}{2}\|\theta_{i_k} - \theta_{i_k}^k\|^2 \leq g_{i_k}(\theta_{i_k}^k; \bar{\boldsymbol{\theta}}_{i_k}^k) - g_{i_k}(\theta_{i_k}^{k+1}; \bar{\boldsymbol{\theta}}_{i_k}^k) + h_{i_k}(\theta_{i_k}^{k+1}; \bar{\boldsymbol{\theta}}_{i_k}^k) - h_{i_k}(\theta_{i_k}^k; \bar{\boldsymbol{\theta}}_{i_k}^k),$$

$$\langle \nabla g_{i_k}(\theta_{i_k}^k; \bar{\boldsymbol{\theta}}_{i_k}^k) - u_{i_k}^k, \theta_{i_k}^k - \theta_{i_k} \rangle - \frac{L+\rho}{2}\|\theta_{i_k} - \theta_{i_k}^k\|^2 \leq f(\boldsymbol{\theta}^k) - f(\boldsymbol{\theta}^{k+1}).$$

$$\text{(B.4)}$$

Let us denote by $\mathbb{E}_{|k}$ the conditional expectation with respect to the random selection of $i_k$, given all the random choices in the previous iterations. Then, we have

$$\mathbb{E}_{|k}\left[\langle \nabla_{i_k} g_{i_k}(\theta_{i_k}^k; \bar{\boldsymbol{\theta}}_{i_k}^k) - u_{i_k}^k, \theta_{i_k}^k - \theta_{i_k} \rangle - \frac{L+\rho}{2}\|\theta_{i_k} - \theta_{i_k}^k\|^2\right]$$
$$= \frac{1}{n}\left(\langle \boldsymbol{\nu}^k, \boldsymbol{\theta}^k - \boldsymbol{\theta} \rangle - \frac{L+\rho}{2}\|\boldsymbol{\theta} - \boldsymbol{\theta}^k\|^2\right),$$

$$\text{(B.5)}$$

for every $\boldsymbol{\nu}^k \in \partial f(\boldsymbol{\theta}^k)$. Using (B.3) we have

$$\langle \boldsymbol{\nu}^k, \, \boldsymbol{\theta}^k - \boldsymbol{\theta} \rangle - \frac{L+\rho}{2}\|\boldsymbol{\theta} - \boldsymbol{\theta}^k\|^2 \leq nf(\boldsymbol{\theta}^k) - n\mathbb{E}_{|k}\left[f(\boldsymbol{\theta}^{k+1})\right],$$

for every $\boldsymbol{\nu}^k \in \partial f(\boldsymbol{\theta}^k)$. Now, we maximize this inequality over $\boldsymbol{\theta} \in \mathcal{X}$ to get

$$\frac{1}{2(L+\rho)}\mathcal{G}^2(\boldsymbol{\theta}^k) \leq nf(\boldsymbol{\theta}^k) - n\mathbb{E}_{|k}\left[f(\boldsymbol{\theta}^{k+1})\right]. \tag{B.6}$$

Now, taking expectation w.r.t. all the iterations we have

$$\mathbb{E}\left[\frac{1}{2(L+\rho)}\mathcal{G}^2(\boldsymbol{\theta}^k)\right] \leq n\mathbb{E}\left[f(\boldsymbol{\theta}^k)\right] - n\mathbb{E}\left[f(\boldsymbol{\theta}^{k+1})\right]. \tag{B.7}$$

Finally, we take the average of this inequality over $k = 1, \dots, K$:

$$\frac{1}{K}\sum_{k=1}^K \mathbb{E}\left[\frac{1}{2(L+\rho)}\mathcal{G}^2(\boldsymbol{\theta}^k)\right] \leq \frac{n}{K}\left(f(\boldsymbol{\theta}^1) - \mathbb{E}\left[f(\boldsymbol{\theta}^{K+1})\right])\right) \leq \frac{n}{K}\left(f(\boldsymbol{\theta}^1) - f^\star\right). \tag{B.8}$$

which concludes the proof.

## B.8 PROOF OF PROPOSITION A.8

Denote $\epsilon_k := \nabla \hat{g}_{i_k}(\theta_{i_k}^k; \bar{\boldsymbol{\theta}}_{i_k}^k) - \hat{u}_{i_k}^k - \left(\nabla g_{i_k}(\theta_{i_k}^k; \bar{\boldsymbol{\theta}}_{i_k}^k) - u_{i_k}^k\right)$. We want to show a low probability for the event $\{t = t_2 < K\} \cup \{t = t_1 < K, t_2 = K\}$. To do so, we prove a low probability for each of these events. For the first event, it is easy to see that the probability of $\{t_2 < K\}$ is

$$\mathbb{P}(t_2 < T) = \mathbb{P}(\bigcup_{k<K} \{\|\epsilon_k\| > F'\}) \leq \sum_{k<K} \mathbb{P}(\{\|\epsilon_k\| > F'\}) \leq \frac{K\sigma^2}{F'^2}. \tag{B.9}$$

Note that we want $\frac{K\sigma^2}{F'^2} \leq \frac{\delta}{4}$ for $0 < \delta < 1$. For the second event, take $k = t$. Then, we have:

$$g_{i_k}(\theta_{i_k}^{k+1}; \bar{\boldsymbol{\theta}}_{i_k}^k) - g^* > G \qquad \|\epsilon_k\| \leq F'$$

which implies $g_{i_k}(\theta_{i_k}^k; \bar{\boldsymbol{\theta}}_{i_k}^k) - g^* \leq G$ due to the $\min\{.\}$ operator. Note that since $t = t_1$, we must have $t_1 < t_2$. This ensures $\|\nabla g_{i_k}(\theta_{i_k}^k; \bar{\boldsymbol{\theta}}_{i_k}^k)\| \leq E$ through Corollary A.11 and boundedness of the update points through Lemma A.6. Now, using Lemma A.7, we get

$$g_{i_k}(\theta_{i_k}^{k+1}; \bar{\boldsymbol{\theta}}_{i_k}^k) - g_{i_k}(\theta_{i_k}^k; \bar{\boldsymbol{\theta}}_{i_k}^k) \leq \langle \nabla g_{i_k}(\theta_{i_k}^k; \bar{\boldsymbol{\theta}}_{i_k}^k), \theta_{i_k}^{k+1} - \theta_{i_k}^k \rangle + \frac{L}{2}\|\theta_{i_k}^{k+1} - \theta_{i_k}^k\|^2$$

$$\leq \|\nabla g_{i_k}(\theta_{i_k}^k; \bar{\boldsymbol{\theta}}_{i_k}^k)\|\|\theta_{i_k}^{k+1} - \boldsymbol{\theta}_{i_k}^k\| + \frac{L}{2}\|\theta_{i_k}^{k+1} - \boldsymbol{\theta}_{i_k}^k\|^2, \tag{B.10}$$

$$\leq E\frac{2}{\rho}\left(E + R + F'\right) + \frac{L}{2}\left[\frac{2}{\rho}\left(E + R + F'\right)\right]^2.$$

Take $F' = {}^{E\rho}/_{9L} - (E + R)$ and note that $E^2 = 2LG$. $F'$ is positive for $\rho \geq {}^{9L(E+R)}/_E$. This is a valid choice of $\rho$ since it satisfies (see Lemma A.7):

$$\rho \geq \frac{2(E + R + F')}{r(E)}.$$

Now, replacing in (B.10) gives:

$$g_{i_k}(\theta_{i_k}^{k+1}; \bar{\boldsymbol{\theta}}_{i_k}^k) - g_{i_k}(\theta_{i_k}^k; \bar{\boldsymbol{\theta}}_{i_k}^k) \leq \frac{G}{2}. \tag{B.11}$$

This means that:

$$g_{i_k}(\theta_{i_k}^k; \bar{\boldsymbol{\theta}}_{i_k}^k) - g^* = g_{i_k}(\theta_{i_k}^k; \bar{\boldsymbol{\theta}}_{i_k}^k) - g_{i_k}(\theta_{i_k}^{k+1}; \bar{\boldsymbol{\theta}}_{i_k}^k) + g_{i_k}(\theta_{i_k}^{k+1}; \bar{\boldsymbol{\theta}}_{i_k}^k) - g^* \geq \frac{G}{2}, \tag{B.12}$$

which essentially implies:

$$\mathbb{P}(\{t_1 < K\} \cap \{t_2 = K\}) \leq \mathbb{P}(g_{i_k}(\theta_{i_k}^k; \bar{\boldsymbol{\theta}}_{i_k}^k) - g^* \geq \frac{G}{2}) \leq \frac{\mathbb{E}[g_{i_k}(\theta_{i_k}^k; \bar{\boldsymbol{\theta}}_{i_k}^k) - g^*]}{\frac{G}{2}}. \qquad \text{(B.13)}$$

Now, we need to calculate $\mathbb{E}[g_{i_k}(\theta_{i_k}^k; \bar{\boldsymbol{\theta}}_{i_k}^k) - g^*]$. Due to (4.7), we have

$$u_{i_k}^k \in \partial h_{i_k}(\theta_{i_k}^k; \bar{\boldsymbol{\theta}}_{i_k}^k) \quad \text{and} \quad \langle \hat{u}_{i_k}^k, \theta_{i_k}^k - \theta_{i_k}^{k+1} \rangle \leq g_{i_k}(\theta_{i_k}^k; \bar{\boldsymbol{\theta}}_{i_k}^k, s^k) - g_{i_k}(\theta_{i_k}^{k+1}; \bar{\boldsymbol{\theta}}_{i_k}^k, s^k) - \frac{\rho}{2} \|\theta_{i_k}^{k+1} - \theta_{i_k}^k\|^2.$$
(B.14)

By adding and subtracting $\langle \hat{u}_{i_k}^k - u_{i_k}^k, \theta_{i_k}^{k+1} - \theta_{i_k}^k \rangle$ and using convexity of $g_{i_k}$, we have

$$\langle u_{i_k}^k, \theta_{i_k}^k - \theta_{i_k}^{k+1} \rangle \leq \langle \hat{u}_{i_k}^k - u_{i_k}^k, \theta_{i_k}^{k+1} - \theta_{i_k}^k \rangle - \langle \nabla \hat{g}_{i_k}(\theta_{i_k}^k; \bar{\boldsymbol{\theta}}_{i_k}^k), \theta_{i_k}^{k+1} - \theta_{i_k}^k \rangle$$
$$- \frac{\rho}{2} \|\theta_{i_k}^{k+1} - \theta_{i_k}^k\|^2,$$
$$\implies \langle \nabla \hat{g}_{i_k}(\theta_{i_k}^k; \bar{\boldsymbol{\theta}}_{i_k}^k), \theta_{i_k}^{k+1} - \theta_{i_k}^k \rangle \leq \langle \nabla \hat{g}_{i_k}(\theta_{i_k}^k; \bar{\boldsymbol{\theta}}_{i_k}^k) - \hat{u}_{i_k}^k - (\nabla g_{i_k}(\theta_{i_k}^k; \bar{\boldsymbol{\theta}}_{i_k}^k) - u_{i_k}^k), \theta_{i_k}^k - \theta_{i_k}^{k+1} \rangle$$
$$- \frac{\rho}{2} \|\theta_{i_k}^{k+1} - \theta_{i_k}^k\|^2 - \langle u_{i_k}^k, \theta_{i_k}^k - \theta_{i_k}^{k+1} \rangle,$$

Since the conditions of Lemma A.7 are satisfied up to time point $k$, we may use the conclusion of this lemma. Therefore, local smoothness of $g_{i_k}$ together with Young's inequality imply:

$$g_{i_k}(\theta_{i_k}^{k+1}; \bar{\boldsymbol{\theta}}_{i_k}^k) - g_{i_k}(\theta_{i_k}^k; \bar{\boldsymbol{\theta}}_{i_k}^k) - \frac{L}{2} \|\theta_{i_k}^{k+1} - \theta_{i_k}^k\|^2 \leq \frac{\rho}{4} \|\theta_{i_k}^{k+1} - \theta_{i_k}^k\|^2$$
$$+ \frac{1}{\rho} \|\nabla \hat{g}_{i_k}(\theta_{i_k}^k; \bar{\boldsymbol{\theta}}_{i_k}^k) - \hat{u}_{i_k}^k - (\nabla g_{i_k}(\theta_{i_k}^k; \bar{\boldsymbol{\theta}}_{i_k}^k) - u_{i_k}^k)\|^2$$
$$- \frac{\rho}{2} \|\theta_{i_k}^{k+1} - \theta_{i_k}^k\|^2 - \langle u_{i_k}^k, \theta_{i_k}^k - \theta_{i_k}^{k+1} \rangle$$
$$= \frac{1}{\rho} \|\nabla \hat{g}_{i_k}(\theta_{i_k}^k; \bar{\boldsymbol{\theta}}_{i_k}^k) - \hat{u}_{i_k}^k - (\nabla g_{i_k}(\theta_{i_k}^k; \bar{\boldsymbol{\theta}}_{i_k}^k) - u_{i_k}^k)\|^2$$
$$- \langle u_{i_k}^k, \theta_{i_k}^k - \theta_{i_k}^{k+1} \rangle - \frac{\rho}{4} \|\theta_{i_k}^{k+1} - \theta_{i_k}^k\|^2.$$

Now, using $\rho \geq \frac{9L(E+R)}{E}$, we know that $\frac{L}{2} \|\theta_{i_k}^{k+1} - \theta_{i_k}^k\|^2 \leq \frac{\rho}{4} \|\theta_{i_k}^{k+1} - \theta_{i_k}^k\|^2$. Therefore, we have:

$$g_{i_k}(\theta_{i_k}^{k+1}; \bar{\boldsymbol{\theta}}_{i_k}^k) - g_{i_k}(\theta_{i_k}^k; \bar{\boldsymbol{\theta}}_{i_k}^k) \leq \frac{1}{\rho} \|\nabla \hat{g}_{i_k}(\theta_{i_k}^k; \bar{\boldsymbol{\theta}}_{i_k}^k) - \hat{u}_{i_k}^k - (\nabla g_{i_k}(\theta_{i_k}^k; \bar{\boldsymbol{\theta}}_{i_k}^k) - u_{i_k}^k)\|^2$$
$$+ h_{i_k}(\theta_{i_k}^{k+1}; \bar{\boldsymbol{\theta}}_{i_k}^k) - h_{i_k}(\theta_{i_k}^k; \bar{\boldsymbol{\theta}}_{i_k}^k).$$

where the last inequality is due to convexity of $h_{i_k}$. Taking expectation with respect to $s \sim \text{Unif}\{1, \ldots, J\}$, summing over iteration number $k$ and using the assumption on boundedness of $h_{i_k}(\theta_{i_k}^k; \bar{\boldsymbol{\theta}}_{i_k}^k) - h_{i_0}(\theta_{i_0}^0; \bar{\boldsymbol{\theta}}_{i_0}^0) \leq H$, we get:

$$g_{i_k}(\theta_{i_k}^{k+1}; \bar{\boldsymbol{\theta}}_{i_k}^k) - g^* \leq g_{i_0}(\theta_{i_0}^0; \bar{\boldsymbol{\theta}}_{i_0}^0) - g^* + \frac{(k+1)\sigma^2}{\rho} + H. \qquad \text{(B.15)}$$

This means that

$$g_{i_k}(\theta_{i_k}^k; \bar{\boldsymbol{\theta}}_{i_k}^k) - g^* \leq g_{i_0}(\theta_{i_0}^0; \bar{\boldsymbol{\theta}}_{i_0}^0) - g^* + \frac{k\sigma^2}{\rho} + H \leq g_{i_0}(\theta_{i_0}^0; \bar{\boldsymbol{\theta}}_{i_0}^0) - g^* + \frac{K\sigma^2}{\rho} + H.$$

By taking $\rho = \Omega(\sqrt{K})$ and $\sigma^2 = \mathcal{O}(1/\sqrt{K})$ we have

$$g_{i_k}(\theta_{i_k}^k; \bar{\boldsymbol{\theta}}_{i_k}^k) - g^* \leq g_{i_0}(\theta_{i_0}^0; \bar{\boldsymbol{\theta}}_{i_0}^0) - g^* + C', \qquad \text{(B.16)}$$

for a constant $C' := \frac{K\sigma^2}{\rho} + H$. Using (B.16) in (B.13) we get

$$\frac{g_{i_k}(\theta_{i_k}^k; \bar{\boldsymbol{\theta}}_{i_k}^k) - g^*}{\frac{G}{2}} \leq \frac{2 (\max_j g_j(\theta_j^0; \bar{\boldsymbol{\theta}}_j^0) - g^* + C')}{G} = \frac{\delta}{4}, \qquad \text{(B.17)}$$

which holds for $G = \frac{8\left(\max_j g_j(\theta_j^0; \bar{\theta}_j^0) - g^* + C'\right)}{\delta}$. Now, replacing in (B.13) gives

$$\mathbb{P}(\{t_1 < K\} \cap \{t_2 = K\}) \leq \mathbb{P}(g_{i_k}(\theta_{i_k}^k; \bar{\theta}_{i_k}^k) - g^* \geq \frac{G}{2}) \leq \frac{\delta}{4}. \tag{B.18}$$

Using $\frac{K\sigma^2}{F'^2} \leq \frac{\delta}{4}$ and $G = \frac{8\left(g_{i_0}(\theta_{i_0}^0; \bar{\theta}_{i_0}^0) - g^* + C'\right)}{\delta}$ we need

$$\frac{K\sigma^2}{(\frac{E\rho}{9L} - (E+R))^2} \leq \frac{\delta}{4}. \tag{B.19}$$

Using $E^2 = 2LG$ and simplifying (B.19), we have:

$$\frac{2G\rho^2}{81L} + (E+R)^2 - \frac{2\rho}{9L}(2LG + ER) \geq \frac{2G\rho^2}{81L} - \frac{2\rho}{9L}(2LG + ER) \geq \frac{4}{\delta}K\sigma^2. \tag{B.20}$$

Replacing $C' = {K\sigma^2}/{\rho} + H$ and the fact that $G\delta \geq 8C'$ by the definition of $G$, gives

$$\rho^2 - \frac{9\rho}{G}(2LG + ER) \geq \frac{\rho(C' - H)(81L)}{4C'}, \tag{B.21}$$

$$\implies \rho \geq 18L + \frac{9ER}{G} + \frac{81L}{4}\left[\frac{C' - H}{C'}\right] \tag{B.22}$$

With this choice of $\rho$ we ensure

$$\mathbb{P}(\{t_1 < K\} \cap \{t_2 = K\}) + \mathbb{P}(\{t_2 < K\}) \leq \delta/2. \tag{B.23}$$

As a result $\mathbb{P}(\{t = K\}) \geq 1 - \delta/2 \geq 1/2$. Using this result we may use the descent Lemma A.7 up to time point $K$. Using the update rule of (4.7), we have:

$$\begin{aligned}
&g_{i_k}(\theta_{i_k}^{k+1}; \bar{\theta}_{i_k}^k, s^k) - h_{i_k}(\theta_{i_k}^{k+1}; \bar{\theta}_{i_k}^k, s^k) \\
&\leq g_{i_k}(\theta_{i_k}^{k+1}; \bar{\theta}_{i_k}^k, s^k) - h_{i_k}(\theta_{i_k}^k; \bar{\theta}_{i_k}^k, s^k) - \langle \hat{u}_{i_k}^k, \theta_{i_k}^{k+1} - \theta_{i_k}^k \rangle \\
&\leq g_{i_k}(\theta_{i_k}^{k+1}; \bar{\theta}_{i_k}^k, s^k) - h_{i_k}(\theta_{i_k}^k; \bar{\theta}_{i_k}^k, s^k) - \langle \hat{u}_{i_k}^k, \theta_{i_k}^{k+1} - \theta_{i_k}^k \rangle + \frac{\rho}{2}\|\theta_{i_k}^{k+1} - \theta_{i_k}^k\|^2 - \frac{\rho}{2}\|\theta_{i_k}^{k+1} - \theta_{i_k}^k\|^2 \\
&\leq g_{i_k}(\theta_{i_k}; \bar{\theta}_{i_k}^k, s^k) - h_{i_k}(\theta_{i_k}^k; \bar{\theta}_{i_k}^k, s^k) - \langle \hat{u}_{i_k}^k, \theta_{i_k} - \theta_{i_k}^k \rangle + \frac{\rho}{2}\|\theta_{i_k} - \theta_{i_k}^k\|^2 - \frac{\rho}{2}\|\theta_{i_k}^{k+1} - \theta_{i_k}^k\|^2,
\end{aligned} \tag{B.24}$$

for any $\boldsymbol{\theta} \in \mathcal{B}(\boldsymbol{\theta}^k, \frac{2}{\rho}\mathcal{G}(\boldsymbol{\theta}^k))$. Now, using Lemma A.7 we have

$$\begin{aligned}
&g_{i_k}(\theta_{i_k}^k; \bar{\theta}_{i_k}^k, s^k) - h_{i_k}(\theta_{i_k}^k; \bar{\theta}_{i_k}^k, s^k) + \langle \nabla\hat{g}_{i_k}(\theta_{i_k}^k; \bar{\theta}_{i_k}^k) - \hat{u}_{i_k}^k, \theta_{i_k} - \theta_{i_k}^k \rangle + \frac{L + \rho}{2}\|\theta_{i_k} - \theta_{i_k}^k\|^2 - \frac{\rho}{2}\|\theta_{i_k}^{k+1} - \theta_{i_k}^k\|^2 \\
&\leq f(\boldsymbol{\theta}^k, s^k) + \langle \nabla\hat{g}_{i_k}(\theta_{i_k}^k; \bar{\theta}_{i_k}^k) - \hat{u}_{i_k}^k - (\nabla g_{i_k}(\theta_{i_k}^k; \bar{\theta}_{i_k}^k) - u_{i_k}^k),\, \theta_{i_k} - \theta_{i_k}^k \rangle + \langle \nabla g_{i_k}(\theta_{i_k}^k; \bar{\theta}_{i_k}^k) - u_{i_k}^k,\, \theta_{i_k} - \theta_{i_k}^k \rangle \\
&\quad + \frac{L + \rho}{2}\|\theta_{i_k} - \theta_{i_k}^k\|^2
\end{aligned} \tag{B.25}$$

Rearranging and using Young's inequality gives

$$\begin{aligned}
&\langle \nabla g_{i_k}(\theta_{i_k}^k; \bar{\theta}_{i_k}^k) - u_{i_k}^k,\, \theta_{i_k}^k - \theta_{i_k} \rangle - \frac{L + \rho}{2}\|\theta_{i_k} - \theta_{i_k}^k\|^2 \leq \\
&f(\boldsymbol{\theta}^k, s^k) - f(\boldsymbol{\theta}^{k+1}, s^k) + \frac{\rho}{4}\|\theta_{i_k} - \theta_{i_k}^k\|^2 + \frac{1}{\rho}\|\nabla\hat{g}_{i_k}(\theta_{i_k}^k; \bar{\theta}_{i_k}^k) - \hat{u}_{i_k}^k - (\nabla g_{i_k}(\theta_{i_k}^k; \bar{\theta}_{i_k}^k) - u_{i_k}^k)\|^2,
\end{aligned} \tag{B.26}$$

which implies:

$$\begin{aligned}
&\langle \nabla g_{i_k}(\theta_{i_k}^k; \bar{\theta}_{i_k}^k) - u_{i_k}^k,\, \theta_{i_k}^k - \theta_{i_k} \rangle - \frac{L + \frac{3}{2}\rho}{2}\|\theta_{i_k} - \theta_{i_k}^k\|^2 \leq \\
&f(\boldsymbol{\theta}^k, s^k) - f(\boldsymbol{\theta}^{k+1}, s^k) + \frac{1}{\rho}\|\nabla\hat{g}_{i_k}(\theta_{i_k}^k; \bar{\theta}_{i_k}^k) - \hat{u}_{i_k}^k - (\nabla g_{i_k}(\theta_{i_k}^k; \bar{\theta}_{i_k}^k) - u_{i_k}^k)\|^2.
\end{aligned} \tag{B.27}$$

Now, taking expectation conditioned on all the information up to iteration $k$ and $t = K$ and also maximizing l.h.s for all $\boldsymbol{\theta} \in \mathcal{B}(\boldsymbol{\theta}^k, \frac{2}{\rho}\mathcal{G}(\boldsymbol{\theta}^k))$, we get:

$$
\mathbb{E}_{s,i_k|k} \left[ \max_{\boldsymbol{\theta} \in \mathcal{B}(\boldsymbol{\theta}^k, \frac{2}{\rho}\mathcal{G}(\boldsymbol{\theta}^k))} \langle \nabla g_{i_k}(\theta_{i_k}^k; \bar{\boldsymbol{\theta}}_{i_k}^k) - u_{i_k}^k, \theta_{i_k}^k - \theta_{i_k} \rangle - \frac{L + \frac{3}{2}\rho}{2} \|\theta_{i_k} - \theta_{i_k}^k\|^2 \right]
$$

$$
\leq \max_{\boldsymbol{\theta} \in \mathcal{B}(\boldsymbol{\theta}^k, \frac{2}{\rho}\mathcal{G}(\boldsymbol{\theta}^k))} \mathbb{E}_{s,i_k|k} \left[ \langle \nabla g_{i_k}(\theta_{i_k}^k; \bar{\boldsymbol{\theta}}_{i_k}^k) - u_{i_k}^k, \theta_{i_k}^k - \theta_{i_k} \rangle - \frac{L + \frac{3}{2}\rho}{2} \|\theta_{i_k} - \theta_{i_k}^k\|^2 \right]
$$

$$
= \frac{1}{n} \max_{\boldsymbol{\theta} \in \mathcal{B}(\boldsymbol{\theta}^k, \frac{2}{\rho}\mathcal{G}(\boldsymbol{\theta}^k))} \mathbb{E}_{s|k} \left[ \langle \boldsymbol{\nu}^k, \boldsymbol{\theta}^k - \boldsymbol{\theta} \rangle - \frac{L + \frac{3}{2}\rho}{2} \|\boldsymbol{\theta} - \boldsymbol{\theta}^k\|^2 \right]
$$

$$
\leq \mathbb{E}_{s,i_k|k} \left[ f(\boldsymbol{\theta}^k, s^k) - f(\boldsymbol{\theta}^{k+1}, s^k) \right] + \frac{1}{\rho} \mathbb{E}_{s,i_k|k} \left[ \|\nabla \hat{g}_{i_k}(\theta_{i_k}^k; \bar{\boldsymbol{\theta}}_{i_k}^k) - \hat{u}_{i_k}^k - (\nabla g_{i_k}(\theta_{i_k}^k; \bar{\boldsymbol{\theta}}_{i_k}^k) - u_{i_k}^k)\|^2 \right]
$$

$$
\leq \mathbb{E}_{s,i_k|k} \left[ f(\boldsymbol{\theta}^k, s^k) - f(\boldsymbol{\theta}^{k+1}, s^k) \right] + \frac{\sigma^2}{\rho},
$$

(B.28)

for every $\boldsymbol{\nu}^k \in \partial f(\boldsymbol{\theta}^k)$. Averaging both hand sides from $k = 0$ to $k = K$ and using $\mathbb{P}(\{t = K\}) \geq 1 - \delta/2 \geq 1/2.$, we have:

$$
\frac{1}{2K} \sum_{k < K} \mathbb{E}_{s,i_k|k} \left[ \max_{\boldsymbol{\theta} \in \mathcal{B}(\boldsymbol{\theta}^k, \frac{2}{\rho}\mathcal{G}(\boldsymbol{\theta}^k))} \langle \nabla g_{i_k}(\theta_{i_k}^k; \bar{\boldsymbol{\theta}}_{i_k}^k) - u_{i_k}^k, \theta_{i_k}^k - \theta_{i_k} \rangle - \frac{L + \frac{3}{2}\rho}{2} \|\theta_{i_k} - \theta_{i_k}^k\|^2 \right]
$$

$$
\leq \frac{\mathbb{P}(\{t = K\})}{K} \sum_{k < K} \mathbb{E}_{s,i_k|k} \left[ \max_{\boldsymbol{\theta} \in \mathcal{B}(\boldsymbol{\theta}^k, \frac{2}{\rho}\mathcal{G}(\boldsymbol{\theta}^k))} \langle \nabla g_{i_k}(\theta_{i_k}^k; \bar{\boldsymbol{\theta}}_{i_k}^k) - u_{i_k}^k, \theta_{i_k}^k - \theta_{i_k} \rangle - \frac{L + \frac{3}{2}\rho}{2} \|\theta_{i_k} - \theta_{i_k}^k\|^2 \Big| t = K \right]
$$

$$
\leq \frac{\mathbb{P}(\{t = K\})}{K} \sum_{k < K} \mathbb{E}_{s,i_k|k} \left[ \max_{\boldsymbol{\theta} \in \mathcal{B}(\boldsymbol{\theta}^k, \frac{2}{\rho}\mathcal{G}(\boldsymbol{\theta}^k))} \langle \nabla g_{i_k}(\theta_{i_k}^k; \bar{\boldsymbol{\theta}}_{i_k}^k) - u_{i_k}^k, \theta_{i_k}^k - \theta_{i_k} \rangle - \frac{L + \frac{3}{2}\rho}{2} \|\theta_{i_k} - \theta_{i_k}^k\|^2 \Big| t = K \right]
$$

$$
\leq \frac{1}{K} \sum_{k < t} \mathbb{E}_{s,i_k|k} \left[ \max_{\boldsymbol{\theta} \in \mathcal{B}(\boldsymbol{\theta}^k, \frac{2}{\rho}\mathcal{G}(\boldsymbol{\theta}^k))} \langle \nabla g_{i_k}(\theta_{i_k}^k; \bar{\boldsymbol{\theta}}_{i_k}^k) - u_{i_k}^k, \theta_{i_k}^k - \theta_{i_k} \rangle - \frac{L + \frac{3}{2}\rho}{2} \|\theta_{i_k} - \theta_{i_k}^k\|^2 \right]
$$

$$
\leq \frac{n}{K} \left[ \mathbb{E} \left[ f(\boldsymbol{\theta}^0) - f(\boldsymbol{\theta}^K) \right] + \frac{K\sigma^2}{\rho} \right] \leq \frac{n}{K} \left[ g_{i_0}(\theta_{i_0}^0; \bar{\boldsymbol{\theta}}_{i_0}^0) - g(\boldsymbol{\theta}^*) + H + \frac{K\sigma^2}{\rho} \right]
$$

$$
= \frac{n}{K} \left[ g_{i_0}(\theta_{i_0}^0; \bar{\boldsymbol{\theta}}_{i_0}^0) - g(\boldsymbol{\theta}^*) + C' \right] = \frac{nG\delta}{8K}.
$$

(B.29)

were in the last equality we used the definition of $G$. Note that the maximum in (B.29) is achieved for $\boldsymbol{\theta} = \boldsymbol{\theta}^k - \frac{1}{L + \frac{3\rho}{2}} \nu_{i_k}^k$ for $\nu_{i_k}^k \in \partial_{i_k} f(\boldsymbol{\theta}^k)$. Since this value is in $\mathcal{B}(\boldsymbol{\theta}^k, \frac{2}{\rho}\mathcal{G}(\boldsymbol{\theta}^k))$, we can replace this value and write:

$$
\frac{1}{2K} \sum_{k < K} \mathbb{E} \left[ \mathcal{G}^2(\boldsymbol{\theta}^k) \right] \leq \frac{(L + \frac{3\rho}{2})nG\delta}{8K}.
$$

(B.30)

Choosing $K \geq \frac{(L + \frac{3\rho}{2})nG}{2\epsilon^2}$ such that we have

$$
\frac{1}{K} \sum_{k < K} \mathbb{E} \left[ \mathcal{G}^2(\boldsymbol{\theta}^k) \right] \leq \frac{(L + \frac{3\rho}{2})nG\delta}{4K} \leq \frac{\delta}{2}\epsilon^2.
$$

(B.31)

Using the fact that $\rho = \Omega(\sqrt{K})$, our convergence guarantee holds for $K = \Omega(1/\epsilon^4)$.

Now, we define the event $\varrho = \left\{ \frac{1}{K} \sum_{k < K} \mathbb{E} \left[ \mathcal{G}^2(\boldsymbol{\theta}^k) \right] > \epsilon^2 \right\}$. Using Markov's inequality we get $\mathbb{P}(\varrho) \leq \delta/2$. Finally, we get $\mathbb{P}(\{t < K\} \cup \varrho) \leq \delta$.

## B.9 PROOF OF LEMMA A.6

Due to (4.7), we have

$$u_{i_k}^k \in \partial h_{i_k}(\theta_{i_k}^k; \bar{\boldsymbol{\theta}}_{i_k}^k) \quad \text{and} \quad \langle \hat{u}_{i_k}^k, \theta_{i_k}^k - \theta_{i_k}^{k+1} \rangle \leq g_{i_k}(\theta_{i_k}^k; \bar{\boldsymbol{\theta}}_{i_k}^k, s^k) - g_{i_k}(\theta_{i_k}^{k+1}; \bar{\boldsymbol{\theta}}_{i_k}^k, s^k) - \frac{\rho}{2} \|\theta_{i_k}^{k+1} - \theta_{i_k}^k\|^2.$$
(B.32)

By adding and subtracting $\langle \hat{u}_{i_k}^k - u_{i_k}^k, \theta_{i_k}^{k+1} - \theta_{i_k}^k \rangle$ and using convexity of $g_{i_k}$, we have

$$\langle u_{i_k}^k, \theta_{i_k}^k - \theta_{i_k}^{k+1} \rangle \leq \langle \hat{u}_{i_k}^k - u_{i_k}^k, \theta_{i_k}^{k+1} - \theta_{i_k}^k \rangle - \langle \nabla \hat{g}_{i_k}(\theta_{i_k}^k; \bar{\boldsymbol{\theta}}_{i_k}^k), \theta_{i_k}^{k+1} - \theta_{i_k}^k \rangle$$
$$- \frac{\rho}{2} \|\theta_{i_k}^{k+1} - \theta_{i_k}^k\|^2,$$
$$\implies \langle u_{i_k}^k - \nabla g_{i_k}(\theta_{i_k}^k; \bar{\boldsymbol{\theta}}_{i_k}^k), \theta_{i_k}^k - \theta_{i_k}^{k+1} \rangle \leq \langle \nabla \hat{g}_{i_k}(\theta_{i_k}^k; \bar{\boldsymbol{\theta}}_{i_k}^k) - \hat{u}_{i_k}^k - \left( \nabla g_{i_k}(\theta_{i_k}^k; \bar{\boldsymbol{\theta}}_{i_k}^k) - u_{i_k}^k \right), \theta_{i_k}^k - \theta_{i_k}^{k+1} \rangle$$
$$- \frac{\rho}{2} \|\theta_{i_k}^{k+1} - \theta_{i_k}^k\|^2,$$

Now, through Cauchy–Schwarz inequality we get

$$\frac{\rho}{2} \|\theta_{i_k}^{k+1} - \theta_{i_k}^k\|^2 \leq \langle \nabla g_{i_k}(\theta_{i_k}^k; \bar{\boldsymbol{\theta}}_{i_k}^k) - u_{i_k}^k, \theta_{i_k}^k - \theta_{i_k}^{k+1} \rangle$$
$$+ \|\nabla \hat{g}_{i_k}(\theta_{i_k}^k; \bar{\boldsymbol{\theta}}_{i_k}^k) - \hat{u}_{i_k}^k - \left( \nabla g_{i_k}(\theta_{i_k}^k; \bar{\boldsymbol{\theta}}_{i_k}^k) - u_{i_k}^k \right) \| \|\theta_{i_k}^{k+1} - \theta_{i_k}^k\|,$$
$$\implies \frac{\rho}{2} \|\theta_{i_k}^{k+1} - \theta_{i_k}^k\|^2 \leq \|\nu_{i_k}^k\| \|\theta_{i_k}^{k+1} - \theta_{i_k}^k\| + \|\hat{u}_{i_k}^k - \nu_{i_k}^k\| \|\theta_{i_k}^{k+1} - \theta_{i_k}^k\|,$$
$$\implies \|\boldsymbol{\theta}^{k+1} - \boldsymbol{\theta}^k\| \leq \frac{2}{\rho} \left( \|\nu_{i_k}^k\| + \|\hat{\nu}_{i_k}^k - \nu_{i_k}^k\| \right),$$
$$\leq \frac{2}{\rho} \left( \|\boldsymbol{\nu}^k\| + \|\hat{\nu}_{i_k}^k - \nu_{i_k}^k\| \right),$$

where $\boldsymbol{\nu}^k \in \partial f(\boldsymbol{\theta}^k), \nu_{i_k}^k \in \partial_{i_k} f(\boldsymbol{\theta}^k), \hat{u}_{i_k}^k \in \partial_{i_k} \hat{f}(\boldsymbol{\theta}^k)$, and the second to last line holds due to the fact that (4.7) updates only the $i_k^{\text{th}}$ block at each iteration. This implies the desired result.

## B.10 PROOF OF LEMMA A.7

By assumption, we know that $g_{i_k}(\theta_{i_k}^k; \bar{\boldsymbol{\theta}}_{i_k}^k) - g^* \leq G$ and $\|\nabla \hat{g}_{i_k}(\theta_{i_k}^k; \bar{\boldsymbol{\theta}}_{i_k}^k) - \hat{u}_{i_k}^k - (\nabla_{i_k} g(\boldsymbol{\theta}^k) - u_{i_k}^k)\| \leq F'$ for $G, F' > 0$. Since this result holds for any $k$, through Corollary A.11 we have $\|\nabla g_{i_k}(\theta_{i_k}^k; \bar{\boldsymbol{\theta}}_{i_k}^k)\| \leq E$. Recall that $g_{i_k}$ is $\ell$-smooth with a subquadratic $\ell$. Using Proposition A.10, $g$ is also $(r, m)$-smooth with $r(u) = \frac{a}{m(u)}$ and $m(u) := \ell(u + a)$ for some $a > 0$. Therefore, we can use Lemma A.9 if we ensure the updates are inside $\mathcal{B}(\boldsymbol{\theta}^k, r(E))$. From the proof of Lemma A.6 (Appendix B.9), we know

$$\|\boldsymbol{\theta}^{k+1} - \boldsymbol{\theta}^k\| \leq \sup_{u \in \partial h_{i_k}(\theta_{i_k}^k; \bar{\boldsymbol{\theta}}_{i_k}^k)} \frac{2(\|\nabla g_{i_k}(\theta_{i_k}^k; \bar{\boldsymbol{\theta}}_{i_k}^k)\| + \|u\| + F')}{\rho} \leq \frac{2(E + R + F')}{\rho}.$$

As a result taking $\rho \geq \frac{2(E+R+F')}{r(E)} = \ell(2E) \frac{2(E+R+F')}{E}$ will satisfy the conditions in Lemma A.9. This implies the desired result.

## B.11 PROOF OF THEOREM A.1

For any $\boldsymbol{\theta}_{i_k} \in \mathcal{M}^{i_k}$, Consider the surrogate function

$$\hat{f}(\theta_{i_k}; \bar{\boldsymbol{\theta}}_{i_k}^k) := g_{i_k}(\theta_{i_k}; \bar{\boldsymbol{\theta}}_{i_k}^k) + r_{i_k}(\theta_{i_k}) - h_{i_k}(\theta_{i_k}^k; \bar{\boldsymbol{\theta}}_{i_k}^k) - \langle u_{i_k}^k, \theta_{i_k} - \theta_{i_k}^k \rangle.$$

where $u_{i_k}^k \in \partial h_{i_k}(\theta_{i_k}; \bar{\boldsymbol{\theta}}_{i_k}^k)$. Also, it is important to mention that $f(\theta_{i_k}^k; \bar{\boldsymbol{\theta}}_{i_k}^k) = f(\boldsymbol{\theta}^k)$ and $\hat{f}(\theta_{i_k}^k; \bar{\boldsymbol{\theta}}_{i_k}^k) = \hat{f}(\boldsymbol{\theta}^k)$. From (A.4) we know that

$$\hat{f}(\boldsymbol{\theta}^{k+1}) \leq \hat{f}(\theta_{i_k}; \bar{\boldsymbol{\theta}}_{i_k}^k) = g_{i_k}(\theta_{i_k}; \bar{\boldsymbol{\theta}}_{i_k}^k) + r_{i_k}(\theta_{i_k}) - h_{i_k}(\theta_{i_k}^k; \bar{\boldsymbol{\theta}}_{i_k}^k) - \langle u_{i_k}^k, \theta_{i_k} - \theta_{i_k}^k \rangle,$$

and further considering the smoothness of $g$, we get

$$\hat{f}(\boldsymbol{\theta}^{k+1}) \le g_{i_k}(\theta_{i_k}^k; \bar{\boldsymbol{\theta}}_{i_k}^k) + \langle \nabla g_{i_k}(\theta_{i_k}^k; \bar{\boldsymbol{\theta}}_{i_k}^k), \theta_{i_k} - \theta_{i_k}^k \rangle + \frac{L}{2}\|\theta_{i_k} - \theta_{i_k}^k\|^2 + r_{i_k}(\theta_{i_k}) - h_{i_k}(\theta_{i_k}^k; \bar{\boldsymbol{\theta}}_{i_k}^k) - \langle u_{i_k}^k, \theta_{i_k} - \theta_{i_k}^k \rangle.$$

By (A.2), we get

$$\hat{f}(\boldsymbol{\theta}^{k+1}) \le f(\boldsymbol{\theta}^k) + \langle \nabla g_{i_k}(\theta_{i_k}^k; \bar{\boldsymbol{\theta}}_{i_k}^k) - u_{i_k}^k, \theta_{i_k} - \theta_{i_k}^k \rangle + \frac{L}{2}\|\theta_{i_k} - \theta_{i_k}^k\|^2 + r_{i_k}(\theta_{i_k}) - r_{i_k}(\theta_{i_k}^k) \tag{B.33}$$

Therefore,

$$\langle \nabla g_{i_k}(\theta_{i_k}^k; \bar{\boldsymbol{\theta}}_{i_k}^k) - u_{i_k}^k, \theta_{i_k}^k - \theta_{i_k} \rangle - r_{i_k}(\theta_{i_k}) + r_{i_k}(\theta_{i_k}^k) \le f(\boldsymbol{\theta}^k) - \hat{f}(\boldsymbol{\theta}^{k+1}) + \frac{L}{2}\|\theta_{i_k} - \theta_{i_k}^k\|^2$$

$$\le g_{i_k}(\theta_{i_k}^k; \bar{\boldsymbol{\theta}}_{i_k}^k) - g_{i_k}(\theta_{i_k}^{k+1}; \bar{\boldsymbol{\theta}}_{i_k}^k) + \frac{L}{2}\|\theta_{i_k} - \theta_{i_k}^k\|^2$$

$$+ \langle u_{i_k}^k, \theta_{i_k}^{k+1} - \theta_{i_k}^k \rangle + r_{i_k}(\theta_{i_k}^k) - r_{i_k}(\theta_{i_k}^{k+1}). \tag{B.34}$$

Therefore, by convexity of $h_{i_k}(\cdot\,; \bar{\boldsymbol{\theta}}_{i_k}^k)$ we have:

$$\langle \nabla g_{i_k}(\theta_{i_k}^k; \bar{\boldsymbol{\theta}}_{i_k}^k) - u_{i_k}^k, \theta_{i_k}^k - \theta_{i_k} \rangle - \frac{L}{2}\|\theta_{i_k} - \theta_{i_k}^k\|^2 - r_{i_k}(\theta_{i_k}) + r_{i_k}(\theta_{i_k}^k) \le g_{i_k}(\theta_{i_k}^k; \bar{\boldsymbol{\theta}}_{i_k}^k) - g_{i_k}(\theta_{i_k}^{k+1}; \bar{\boldsymbol{\theta}}_{i_k}^k)$$

$$+ h_{i_k}(\theta_{i_k}^{k+1}; \bar{\boldsymbol{\theta}}_{i_k}^k) - h_{i_k}(\theta_{i_k}^k; \bar{\boldsymbol{\theta}}_{i_k}^k) + r_{i_k}(\theta_{i_k}^k) - r_{i_k}(\theta_{i_k}^{k+1}),$$

$$\langle \nabla g_{i_k}(\theta_{i_k}^k; \bar{\boldsymbol{\theta}}_{i_k}^k) - u_{i_k}^k, \theta_{i_k}^k - \theta_{i_k} \rangle - \frac{L}{2}\|\theta_{i_k} - \theta_{i_k}^k\|^2 - r_{i_k}(\theta_{i_k}) + r_{i_k}(\theta_{i_k}^k) \le f(\boldsymbol{\theta}^k) - f(\boldsymbol{\theta}^{k+1}). \tag{B.35}$$

Let us denote by $\mathbb{E}_{|k}$ the conditional expectation with respect to the random selection of $i_k$, given all the random choices in the previous iterations. Then, we have

$$\mathbb{E}_{|k}\left[\langle \nabla g_{i_k}(\theta_{i_k}^k; \bar{\boldsymbol{\theta}}_{i_k}^k) - u_{i_k}^k, \theta_{i_k}^k - \theta_{i_k} \rangle - \frac{L}{2}\|\theta_{i_k} - \theta_{i_k}^k\|^2 - r_{i_k}(\theta_{i_k}) + r_{i_k}(\theta_{i_k}^k)\right]$$

$$= \frac{1}{n}\sum_{i=1}^n \left(\langle \nabla g_i(\theta_i^k; \bar{\boldsymbol{\theta}}_i^k) - u_i^k, \theta_i^k - \theta_i \rangle - \frac{L}{2}\|\theta_i - \theta_i^k\|^2 - r_i(\theta_i) + r_i(\theta_i^k)\right) \tag{B.36}$$

$$= \frac{1}{n}\left(\langle \boldsymbol{\nu}^k, \boldsymbol{\theta}^k - \boldsymbol{\theta} \rangle - \frac{L}{2}\|\boldsymbol{\theta} - \boldsymbol{\theta}^k\|^2 - r(\boldsymbol{\theta}) + r(\boldsymbol{\theta}^k)\right),$$

for every $\boldsymbol{\nu}^k \in \partial f(\boldsymbol{\theta}^k)$. We have used the block separability of the function $r(\boldsymbol{\theta})$. Using (B.36) we have

$$\langle \boldsymbol{\nu}^k, \boldsymbol{\theta}^k - \boldsymbol{\theta} \rangle - \frac{L}{2}\|\boldsymbol{\theta} - \boldsymbol{\theta}^k\|^2 - r(\boldsymbol{\theta}) + r(\boldsymbol{\theta}^k) \le nf(\boldsymbol{\theta}^k) - n\mathbb{E}_{|k}\left[f(\boldsymbol{\theta}^{k+1})\right].$$

Now, we maximize this inequality over $\boldsymbol{\theta} \in \mathcal{M}$ to get

$$\mathrm{gap}_{\mathcal{M}}^L(\boldsymbol{\theta}^k) \le nf(\boldsymbol{\theta}^k) - n\mathbb{E}_{|k}\left[f(\boldsymbol{\theta}^{k+1})\right]. \tag{B.37}$$

Now, taking expectation w.r.t. all the iterations we have

$$\mathbb{E}\left[\mathrm{gap}_{\mathcal{M}}^L(\boldsymbol{\theta}^k)\right] \le n\mathbb{E}\left[f(\boldsymbol{\theta}^k)\right] - n\mathbb{E}\left[f(\boldsymbol{\theta}^{k+1})\right]. \tag{B.38}$$

Finally, we take the average of this inequality over $k = 1, \dots, K$:

$$\frac{1}{K}\sum_{k=1}^K \mathbb{E}\left[\mathrm{gap}_{\mathcal{M}}^L(\boldsymbol{\theta}^k)\right] \le \frac{n}{K}\left(f(\boldsymbol{\theta}^1) - \mathbb{E}\left[f(\boldsymbol{\theta}^{K+1})\right]\right) \le \frac{n}{K}\left(f(\boldsymbol{\theta}^1) - f^\star\right). \tag{B.39}$$

We complete the proof by noting that the minimum of $\mathrm{gap}_{\mathcal{M}}^L(\boldsymbol{\theta}^k)$ over $k = 1, \dots, K$ is smaller than or equal to the average gap.