# OpenReview forum: "The Multi-Block DC Function Class: Theory, Algorithms, and Applications"
_ICLR.cc/2026/Conference — ICLR 2026 Conference Withdrawn Submission_

### Official Review · Reviewer_uPqw · 2025-10-28

**Soundness:** 3
**Presentation:** 3
**Contribution:** 3
**Rating:** 6
**Confidence:** 3

**Summary:**

This paper introduces the Multi-Block DC (BDC) function class, a generalization of difference-of-convex (DC) programming to functions that admit DC decompositions per parameter block. The authors provide explicit, constructive BDC formulations for practical problems, including deep ReLU networks (with extensions to MSE regression and cross-entropy classification losses). The paper further develops the Block DC algorithms, including proximal and stochastic versions, with non-asymptotic convergence guarantees. Empirical performance is also illustrated through experiments.

**Strengths:**

1. The BDC class extends DC programming to multi-block settings, addressing practical challenges in finding global DC decompositions. The BDC definition is simple yet powerful.

2. The paper provides explicit tools and examples for formulating problems as BDC, making it accessible for applications in machine learning (e.g., tensor decomposition, neural network training).

3. The proposed BDCA algorithms, including stochastic extensions, come with rigorous non-asymptotic convergence rates, which are solid and clearly stated.

**Weaknesses:**

1. While the abstract mentions "several experiments", the paper focuses heavily on theory and algorithms. The current results compare mainly to vanilla SGD and on modest datasets. Full evaluation (e.g., benchmarks against baselines like ADAM on real datasets) is not enough, potentially weakening claims of practical superiority.

2. For the deep ReLU BDC formulation, how does it handle common extensions like batch normalization, dropout, or other activations (e.g., GELU)? Is there a way to extend the constructive approach to these?

3. The paper should also talk about computation and memory overhead of the proposed methods versus standard training.

**Questions:**

See the Weakness part.

---

> ### Author Response · Authors · 2025-11-20
>
> We thank the reviewer for their positive evaluation. Below, please find our responses to the raised weaknesses.
>
> ---
>
> **Weakness 1**
>
> We agree with the reviewer that the phrase “several experiments” in the abstract should be toned down. We will revise the final sentence to:
>
> > “…demonstrate broad applicability of our method through several applications.”
>
> As the reviewer notes, our core contribution is the identification of a well-structured class (multi-block DC functions) and a block-proximal-DC algorithm with non-asymptotic convergence guarantees for this class. The ReLU network example demonstrates that widely used architectures naturally fit this structure. Our proof-of-concept experiments show that block-proximal-DC updates behave differently from SGD: they consistently reduce overfitting while achieving comparable test error (see the new experiments).
>
> We also expect BDC algorithms to overfit less because they rely on convex, structure-preserving approximations rather than the linear approximations used in GD. A related example is the EM algorithm, which can be viewed as a DCA for exponential-family likelihoods and often outperforms GD. This suggests that DC-aware updates induce different optimization dynamics, motivating further study beyond the scope of this work.
>
> To further strengthen the empirical section, we added two new experiments: (i) a comparison with GD on the SDL task, and (ii) an optimization parameter tuning study of both SGD and SBDCA on the neural network training problem. Due to space constraints, the SDL results are included in Section A.6.2 (Figure 4), and the updated neural network results appear in Figure 3. In the neural network experiments, the revised plots show that BDC-Prox reduces overfitting while maintaining comparable test accuracy. In the SDL experiments, BDCA converges faster and reaches a lower objective value than GD.
>
> ---
>
> **Weakness 2**
>
> Below, we discuss how our model deals with each one of the raised extensions.
>
> *Dropout:*
>
> Dropout multiplies activations by a random 0–1 mask. For any fixed mask, this operation is simply a coordinatewise linear map, which does not interfere with the blockwise DC structure used by our BDC recursion. For every realization of the dropout mask, the ReLU-based BDC structure remains unchanged, and the loss can still be written as a difference of blockwise convex functions. Since the mask is independent of the parameters, the expected objective under dropout also remains DC with blockwise convex components. In this sense, our decomposition is fully compatible with dropout.
>
> *Batch normalization:*
>
> Batch normalization can be viewed as an affine map applied to activations, followed by a learned scale and shift. From the BDC perspective, this affine transformation can be absorbed into the surrounding linear layers. Affine maps preserve the structure we rely on and do not break blockwise convexity of the DC components when incorporated into the corresponding affine block. Our formal results are stated for affine–ReLU layers, and the same constructive BDC idea algebraically extends to architectures where batch normalization layers are folded into the linear part.
>
> *Other activations (e.g., GELU, sigmoid, tanh):*
>
> Many common activation functions admit a DC decomposition at the scalar level. However, our deep-network BDC construction relies on specific properties of ReLU (namely piecewise linearity, separability across coordinates, and max-based identities) used throughout the recursion to propagate the BDC structure layer by layer. For more general activations such as GELU, sigmoid, or tanh, this simple piecewise-linear structure is no longer available, and obtaining an explicit multi-block DC decomposition of the full training loss is not straightforward. Extending the constructive BDC approach beyond ReLU is therefore a natural and promising direction for future work.
>
> ---
>
> **Weakness 3**
>
> We assume the reviewer is referring to training ReLU networks. The computation and memory requirements of our method remain very close to standard training. The BDC decomposition is built once at initialization, contains only a small fixed number of terms independent of depth, and we never construct the symbolic expansion that exists only in theory. For more explanation, see our response to question 1 of reviewer SV7Z.
>
> Each training step evaluates the BDC components using the same basic operations as a standard forward pass, matrix multiplications, ReLUs, and elementwise max, so the computational cost is within a small constant factor of ordinary training. Memory usage is also similar, since we store only the usual quantities needed for gradient computation.
>
> As noted in Remark 5.1, the stochastic blockwise update computes gradients only for the selected random layer, so backpropagation runs only up to that layer rather than through the entire network. This reduces the gradient cost and can make each update cheaper than full backpropagation.

---

### Official Review · Reviewer_qDo4 · 2025-10-29

**Soundness:** 2
**Presentation:** 3
**Contribution:** 2
**Rating:** 4
**Confidence:** 4

**Summary:**

In this paper, the authors study a class of (multi-)block DC programs that is **provably** broader than the classical class of DC programs. Specifically, given an objective function $f(\theta)$ with $\theta \in \mathbb{R}^d$, it is said to be a multi-block DC function if there exists a partition of the $d$ coordinates into $n$ blocks such that $f$ admits a DC decomposition with respect to each block of coordinates when the remaining blocks are fixed.

The main motivation stems from the observation that identifying a block DC structure is often much easier than finding a full DC structure, especially in problems with coupled variables. The authors further demonstrate that for monomials, an exponential number of atoms is required to represent them as DC functions, whereas only a polynomial number of atoms suffices under the block DC formulation.

Exploiting this block DC structure, the authors propose a block DC algorithm, which combines the principles of DCA with randomized block selection. Under the assumption that the first DC components (in each block) are $L$-smooth (or generalized $L$-smooth), the squared gradient norm of the objective function---or its expected value in the stochastic case---converges to zero at a rate of $\mathcal{O}(1/K)$, matching the known rate for both stochastic and deterministic DCA.

**Strengths:**

- The proposed framework is quite general and practically relevant. In many applications, it is indeed much easier to identify a block DC structure than a full DC structure, particularly when the variables are coupled. The presentation of the block DC formulation for neural networks with ReLU activation is also neat compared to existing approaches that rely on full-form DC decompositions for such composite functions. **P.S.** Please also do take a look at the paper *Cui, Y., He, Z., & Pang, J. S. (2020). Multicomposite nonconvex optimization for training deep neural networks. SIAM Journal on Optimization, 30(2), 1693-1723.* that gives explicit DC/MM structure for ReLU-activated neural networks of arbitrary depth.

- The observation that DC decompositions of monomials exhibit exponential complexity, in contrast to the polynomial complexity of BDC, is insightful and provides solid motivation for the paper.

- The paper is generally well-written.

**Weaknesses:**

- The novelty of the paper appears quite limited. The core algorithmic design was previously introduced by [Pham et al., 2022] for DC functions (albeit not for BDC), and this prior work is not cited. Given this, the main contribution of the present paper likely lies in the non-asymptotic convergence analysis (for a larger class of BDC), which provides a rate comparable to that of standard (stochastic) DCA. If this is indeed the key advancement, the authors should explicitly emphasize this analytical contribution and highlight the technical contribution to obtain this results.

- Another major concern is that the experiments do not show the advantages of the proposed method. The BDCA's performance is actually worse than simple SGD in all cases. Also, the classification accuracies of 90% for MNIST and 50% for CIFAR10 are very far from SOTA results. This makes me wonder if the method really works for complicated tasks.

**REFERENCE**

Pham, V. T., Luu, H. P. H., & Le Thi, H. A. (2022). A block coordinate DCA approach for large-scale kernel SVM. In International Conference on Computational Collective Intelligence (pp. 334-347). Cham: Springer International Publishing.

**Questions:**

- The definition of the gradient norm $\mathcal{G}$ seems invalid to me. The notion of $\partial f$ is ill-posed for DC functions, what kind of $\partial$ is it?

---

> ### Author Response · Authors · 2025-11-20
>
> We thank the reviewer for their evaluation. Below, please find our responses to the raised weaknesses and questions.
>
> ---
>
> **Weakness 1**
>
> We respectfully disagree that the novelty is limited. Our core contributions are the followings:
>
> 1. We identify and motivate a new well-structured class of functions, namely multi-block DC (BDC) functions.
>
> 2. We study fundamental properties of this class, including an exponential separation in representation complexity compared to standard DC formulations, and we present a new representation for deep ReLU networks.
>
> 3. We propose a multi-block variant of the DC algorithm designed specifically for BDC functions, together with non-asymptotic convergence guarantees. We further extend this algorithm to a relaxed smoothness assumption (generalized smoothness) and to the stochastic setting.
>
> 4. We demonstrate the applicability of this function class and our method across several problems.
>
> The reviewer focused solely on item 3 above, and even within that component, their interpretation understates the scope of the contribution. As the reviewer acknowledges, our algorithmic development goes beyond the discussion in [Pham et al., 2022]. Our setting requires weaker structural assumptions: we work with BDC structure rather than DC, and our analysis establishes non-asymptotic convergence rates under generalized smoothness and for both batch and stochastic gradients. These aspects are not addressed in [Pham et al., 2022]. To the best of our knowledge, no prior work (even within DC optimization) has studied BDC/DC algorithms under the generalized smoothness assumption. Our probabilistic convergence analysis is new for both the batch and stochastic cases.
>
> Finally, our work discusses in detail the more relevant prior paper, [Maskan et al., 2024]. The differences between our contributions and theirs are clearly explained in the related work section (see lines 118–124).
>
> ---
>
> **Weakness 2**
>
> The ReLU-network example illustrates that widely used architectures naturally fit the proposed multi-block DC structure and therefore provide a meaningful testbed for our approach. Our proof-of-concept experiments show that block-proximal-DC updates behave differently from SGD: they consistently reduce overfitting while maintaining similar generalization performance (see the new experiments). We expect BDC to overfit less because they rely on convex, structure-preserving approximations rather than the linear approximations used in GD. DC-aware updates induce different optimization dynamics, motivating further study beyond the scope of this work.
>
> In this paper we lay the foundation for studying and motivating the BDC function class. We first demonstrate its expressivity and then propose an algorithm tailored to this structure. The application section shows several problems that naturally fall within this class and can be addressed using our method with performance comparable to existing techniques. Based on our BDC formulation of the ReLU networks, we evaluated the performance of our algorithm on these networks. Clearly, we do not expect SOTA-level accuracy on image datasets (e.g. CIFAR10) with these networks.
>
> To further strengthen the empirical section, we added two new experiments: (i) a comparison with GD on the SDL task, and (ii) an optimization parameter tuning study of both SGD and SBDCA on the neural network training problem. Due to space constraints, the SDL results are included in Section A.6.2 (Figure 4), and the updated neural network results appear in Figure 3. In the neural network experiments, the revised plots show that BDC-Prox reduces overfitting while maintaining comparable test accuracy. In the SDL experiments, BDCA converges faster and reaches a lower objective value than GD.
>
> ---
>
> **Question**
>
> Thank you for pointing this out. We defined this quantity in the Appendix (line 691), but we mistakenly omitted it from the main body. The intended definition is
> $$
> \partial f(\boldsymbol{\theta}) = \nabla g_i(\theta_i; \bar{\boldsymbol{\theta}}_i) - u_i,
> \qquad
> u_i \in \partial h_i(\theta_i; \bar{\boldsymbol{\theta}}_i),
> $$
> We will add this explicit definition to the main text to avoid ambiguity.

---

### Official Review · Reviewer_SV7Z · 2025-10-30

**Soundness:** 4
**Presentation:** 3
**Contribution:** 3
**Rating:** 8
**Confidence:** 2

**Summary:**

The paper introduces multi-block DC (BDC) functions, a special class of nonconvex functions. Each function is split into blocks of variables, and within each block, it can be written as a convex function minus another convex function. This allows some functions, such as monomials or deep ReLU networks, to be represented more compactly than with standard DC decomposition.

The authors show how to decompose deep ReLU networks into BDC form, develop algorithms to minimize BDC functions (including a stochastic block method), and prove convergence guarantees. They also provide numerical experiments to illustrate the practicality of the decomposition for analyzing neural networks.

**Strengths:**

The paper is conceptually interesting and clearly demonstrates an advantage of the BDC formulation over classical DC decomposition, particularly in settings such as the training of deep ReLU networks. The authors provide convergence guarantees for their proposed algorithms, which adds theoretical rigor to the work. In addition, the paper validates its contributions with numerical experiments on relevant examples, showing that the proposed methods are practical.

**Weaknesses:**

The encoding size of the BDC decomposition for deep ReLU networks is not clearly discussed, and it seems that there could be an exponential blow-up with the number of layers. It would be helpful if the authors could clarify this point. Furthermore, the theoretical convergence guarantees assume $L_i$-smoothness of the functions, which does not hold for ReLU networks. A discussion of this limitation and its implications for practical use would strengthen the paper.

**Questions:**

- Can the authors provide an estimate of the encoding size or complexity of the BDC decomposition for deep ReLU networks, and discuss - In the paper https://arxiv.org/pdf/2411.03006, Proposition 4.2 presents a more efficient way to decompose a ReLU function into a difference of two convex functions (using maxout, but this should be adaptable to ReLU). Can this approach be used to obtain a smaller BDC decomposition of the training problem?
- How do the assumptions in the convergence proofs (e.g., $L_i$-smoothness) affect the applicability of the algorithms to ReLU networks?

---

> ### Author Response · Authors · 2025-11-20
>
> We thank the reviewer for their positive evaluation. Below, please find our responses to the raised questions.
>
> ---
>
> **Question 1**
>
> *Encoding size:*
>
> Our BDC recursion applies nonnegative linear maps and coordinatewise max operations. If one were to fully expand the resulting expressions symbolically, each max doubles the number of affine branches. After $ L-1 $ hidden layers, each coordinate of $ (Z_{L-1}^+, Z_{L-1}^-) $ becomes a maximum of up to $ 2^{L-1} $ affine functions. With widths $ n_1,\dots,n_L $, the worst-case symbolic encoding size is
> $$
> O\left(\sum_{l=1}^L n_l  2^{l-1}\right).
> $$
> This is linear in width and exponential in depth. This exponential behavior concerns only the **symbolic** representation. It does not affect runtime because:
>
> (i) the BDC recursion uses the same operations as a standard forward pass (matrix multiplications, ReLUs, elementwise max),
>
> (ii) we never construct the symbolic expansion, and
>
> (iii) the BDC decomposition is computed once and reused during training.
>
> Also, the functional form
> $$
> F_x(\theta) = A_x(\theta) - B_x(\theta)
> $$
> always contains exactly six summands, independent of depth; depth only increases the internal nesting of max operations.
>
>
> *Relation to Proposition 4.2 (Hertrich & Loho, 2025):*
>
> The construction in Proposition 4.2 is conceptually related to our layerwise splitting, but it cannot yield a smaller BDC decomposition of the **training problem** for two reasons:
>
> 1. **Function-level vs. parameter-level decomposition.**
>    Proposition 4.2 gives, for each fixed $\theta$, a pair $(g_\theta, h_\theta)$ satisfying
>    $$
>    F_x(\theta) = g_\theta(x) - h_\theta(x).
>    $$
>    BDC requires a **single** decomposition
>    $$
>    F_x(\theta) = A(\theta) - B(\theta)
>    $$
>    that holds for **all** $\theta$ and is convex in each parameter block. Proposition 4.2 produces a different pair $(g_\theta, h_\theta)$ for each $\theta$, so it cannot be used as a global BDC split.
>
> 2. **Loss of convexity under reparameterization.**
>    The monotone networks in Proposition 4.2 are convex in their **own** parameters, but these parameters are obtained from $\theta$ via a non-affine, piecewise-linear transformation (splitting $w = w_+ - w_-$ and propagating negative parts across layers).
>    When $(g_\theta, h_\theta)$ are re-expressed in terms of the original $\theta$, blockwise convexity is not preserved. Hence, the resulting functions cannot satisfy the BDC convexity conditions.
>
> In summary, Proposition 4.2 provides a decomposition at the **function level** for a fixed network, while BDC requires a **parameter-wise** decomposition valid for the entire model class. Therefore, it cannot produce a smaller or valid BDC decomposition of the training objective.
>
> ---
>
> **Question 2**
>
> ReLU neural networks are indeed not $L_i$-smooth. However, what we show in the paper is that the first component of the DC decomposition in (3.2) satisfies a local bound on its gradient norm. This behavior is illustrated in Figure 1 and further discussed in Appendix A.6.1. For this reason, we analyze the algorithm under a weaker assumption: the first BDC component satisfies a generalized smoothness condition, which is fully aligned with the structure of ReLU networks. The other assumptions used in the convergence proof remain valid for ReLU models.

---

### Official Review · Reviewer_xn9a · 2025-11-01

**Soundness:** 3
**Presentation:** 2
**Contribution:** 2
**Rating:** 4
**Confidence:** 3

**Summary:**

This paper introduces the Multi-Block DC (BDC) class—a broad new family of structured nonconvex functions.

The authors show that this block-wise structure is more general and powerful than the classical DC class.

They provide explicit, constructive BDC decompositions for modern machine learning models, including deep ReLU networks.

They also establish non-asymptotic convergence guarantees under batch, stochastic, and generalized smoothness settings.

The method is shown to be applicable to tasks such as sparse dictionary learning, multitask feature learning, and neural network training.

**Strengths:**

1. This paper shows that deep ReLU networks can be reformulated as a non-smooth DC function, as presented in Equation (3.2).

2. This paper considers a suite of algorithms—batch, stochastic, and proximal—with detailed non-asymptotic convergence analyses under various conditions.

**Weaknesses:**

1. The authors reformulate ReLU neural networks as equivalent DC optimization problems and solve them using proximal DC algorithms. However, the motivation behind this reformulation and the choice of the proximal DC approach is not clearly justified. It is also unclear what advantages these methods offer over simpler and widely used algorithms such as SGD.

2. The proximal DC algorithm and its theoretical analysis lack novelty. Many of the results do not appear particularly new. The algorithm discussed in Section 4 is a well-established method—essentially a form of coordinate descent—that has been extensively used for various nonconvex optimization problems (including those listed in Section 5). Hence, both the algorithm and the analysis feel incremental.

3. While the paper presents numerical experiments, the empirical section is rather limited compared to the theoretical development. The experiments mainly serve as proof-of-concept demonstrations rather than comprehensive validations against state-of-the-art methods.

4. The proposed algorithms introduce considerable complexity, as each block update involves solving a convex subproblem. The computational cost of these inner-loop optimizations—particularly for complex blocks such as neural network layers—is not thoroughly discussed or compared with simpler alternatives like SGD.

5. Sections 3 and 5 feel somewhat disconnected; the applications are presented in a fragmented manner rather than being integrated into a cohesive narrative.

6. It remains unclear why the ReLU network is assumed to satisfy the (L_i)-smoothness condition.

**Questions:**

NA

---

> ### Author Response · Authors · 2025-11-20
>
> We thank the reviewer for their comments. Below, please find our responses to the raised weaknesses.
>
> ---
>
> **Weakness 1**
>
> Our main contribution is the identification of a well-structured class, namely multi-block DC functions, together with a block-proximal DC algorithm that has non-asymptotic convergence guarantees for this class. The ReLU-network example shows that widely used architectures naturally fall into this class, which highlights the relevance of the framework and its potential for further development. Our proof-of-concept experiments also show that block-proximal DC updates behave differently from SGD: they consistently reduce overfitting while maintaining similar test error (see the new experiments).
>
> We also expect BDC algorithms to overfit less because they rely on convex, structure-preserving approximations rather than the linear approximations used in GD. DC-aware updates induce different optimization dynamics, motivating further study beyond the scope of this work.
>
> Finally, our choice to use a proximal-type BDC algorithm is motivated by the fact that neural-network objectives do not satisfy the global Lipschitz-gradient assumption. Therefore, we used the notion of generalized smoothness for BDC functions. Under this assumption, a proximal BDC algorithm is necessary to guarantee convergence. This point is explained in lines 272–282 and 299–303.
>
> ---
>
> **Weakness 2**
>
> We respectfully disagree with the reviewer’s assessment regarding our algorithm form and analysis.
>
> *Algorithm Form:*
>
> The reviewer’s concern is based on a surface-level similarity between our update rule and proximal coordinate descent. However, the key novelty is not the update form but the function class to which it applies, namely the Multi-Block DC (BDC) decomposition. The BDC-Prox algorithm exploits this new structure, which ensures that all subproblems remain convex by construction. Note that the proximal term is required under generalized smoothness and stochasticity. This behavior is structurally induced by the BDC formulation and is not a design choice that aims to imitate coordinate-type methods.
>
> There are important structural differences between our BDC algorithm and classical coordinate descent, as emphasized in lines 118–123. Our algorithm is the first instantiation of optimization methods that explicitly exploit BDC structure, which classical DC or Block-Coordinate DCA methods [Maskan et al. 2024] cannot represent.
>
> *Algorithmic Analysis:*
>
> Our analysis uses the generalized smoothness assumption. To the best of our knowledge, *no prior work* (even within DC optimization) has studied BDC or DC algorithms under this assumption. We provide a probabilistic convergence analysis for both batch and stochastic gradients for the first time. The proximal term is essential for two reasons:
>
> (i) Under generalized smoothness, we only assume local smoothness, so each step must remain sufficiently close to the previous iterate for the assumption to hold.
>
> (ii) In the stochastic setting, the proximal term controls the distance between updates as a function of the variance and the proximity parameter.
>
> Maskan, Hoomaan, et al. “Block Coordinate DC Programming.” arXiv preprint arXiv:2411.11664 (2024).
>
> ---
>
> **Weakness 3**
>
> We agree that the focus of the paper is more theoretical. The goal of the empirical section is therefore to validate the structural and algorithmic claims, establishing the breadth and relevance of the proposed framework. We showed that a wide range of problems naturally fit into this class and can be addressed using our algorithm with performance comparable to existing methods.
>
> To further strengthen the paper, we added two new experiments in the revision: (i) a comparison with GD on the sparse dictionary learning task (SDL), and (ii) an optimization parameter tuning study of both SGD and SBDCA on the neural network training problem. Due to space constraints, the SDL results are included in Section A.6.2 (Figure 4), and the updated neural network results appear in Figure 3. In the neural network experiments, the revised plots show that BDC-Prox reduces overfitting while maintaining comparable test accuracy. In the SDL experiments, BDCA converges faster and reaches a lower objective value than GD.

---

> > ### Author Response · Authors · 2025-11-20
> >
> > **Weakness 4**
> >
> > Most of the foundational literature in DC programming analyzes convergence in terms of the “main loop” of the algorithm (Yurtsever and Sra, 2022; Abbaszadehpeivasti et al., 2023; Pham Dinh et al., 2022, references are from the paper). This perspective comes from the traditional optimization model where an oracle such as “solving a smooth convex problem exactly” is assumed to run in unit time, or at least at a cost that does not scale with the number of outer-loop iterations. Under this model, the dominant complexity measure is how many times the algorithm calls this oracle (i.e., the number of DCA iterations), rather than the low-level cost of solving each subproblem. This viewpoint is also consistent with the generality of BDC algorithms: different BDC decompositions for the same problem can lead to subproblems with very different computational costs.
> >
> > We emphasize that in each BDC iteration we update only one block, and that block subproblem is convex by construction. Because of this block structure, the per-iteration computational cost is not directly comparable to the cost of an SGD step, and a strict one-to-one comparison of inner-loop complexity is not straightforward.
> >
> > ---
> >
> > **Weakness 5**
> >
> > In Section 3.1, we show that BDC decompositions for monomials are exponentially smaller than standard DC decompositions, and we directly use this fact in the SDL experiment in Section 5. In the SDL setting, the term $XD$ has a quadratic DC decomposition, whereas its BDC decomposition naturally separates into the two blocks $X$ and $D$. Another example of this type is the tensor decomposition case mentioned in the Section 2.
> >
> > In addition, the same BDC decomposition of neural networks discussed in Section 3.2 is empirically validated in Section 5, which shows that the theoretical developments and the applications fit into a consistent framework.
> >
> > ---
> >
> > **Weakness 6**
> >
> > ReLU neural networks are indeed not $L_i$-smooth. However, what we show in the paper is that the first component of the DC decomposition in (3.2) satisfies a local bound on its gradient norm. This behavior is illustrated in Figure 1 and further discussed in Appendix A.6.1. For this reason, we analyze the algorithm under the weaker assumption that the first BDC component satisfies a generalized smoothness condition. This assumption is fully aligned with the structure of ReLU networks and accurately reflects the properties of the decomposition used in our analysis.

---

### Note · Authors · 2026-05-03

**Comment:**

It was later submitted to other venue.

**Withdrawal Confirmation:**

I have read and agree with the venue's withdrawal policy on behalf of myself and my co-authors.

---

### Meta-Review · Area_Chair_DcNM · 2026-01-06

**Summary:**

Disagreement is whether the algorithmic component and analysis are sufficiently novel relative to established DC/DCA and block-coordinate variants, and whether the experiments demonstrate practical value beyond proof-of-concept. The rebuttal clarifies several technical points and adds experiments, but does not fully resolve concerns in my opinion. Given the mixed scores 8, 6, 4, 4 and the remaining gaps, I recommend rejection.

**Reviewer Concerns:**

Novelty. Multiple reviewers questioned whether the proposed methods are largely adaptations of prior DC/DCA or block-coordinate DC work, with missing or insufficiently emphasized related work and unclear delineation of what is fundamentally new.

Motivation and practical advantage. It's unclear why the reformulation and block-prox updates are preferable to standard optimizers (e.g., SGD/Adam) for the stated ML applications.

Limited experiments. Weak or incomplete baselines and results that do not convincingly support performance or usability claims for modern ML tasks.

Details. There were concerns about the cost of solving convex subproblems per block and lack of quantitative runtime/memory comparisons. Also, about smoothness assumptions for ReLU settings and ambiguity around the stationarity/gradient-like measure.

**Reviewer Scores:**

Two "4"s (xn9a and qDo4) likely soften, but unclear by how much.

---

### Decision · Program_Chairs · 2026-01-26

Reject